



**Real-Time Aerosol Optical Properties, Morphology and Mixing**
**States under Clear, Haze and Fog Episodes in the Summer of Urban**
**Beijing**
Rui Li[1], Yunjie Hu[1], Ling Li[1], Hongbo Fu[1,2,*], Jianmin Chen[1,*]
[1] *Shanghai Key Laboratory of Atmospheric Particle Pollution and Prevention, Department of*
*Environmental Science & Engineering, Fudan University, Shanghai 200433, China Chinese*
*Academy of Sciences, Institute of Atmosphere Physics, Beijing 100029.*
[2] *Collaborative Innovation Center of Atmospheric Environment and Equipment Technology*
*(CICAEET), Nanjing University of Information Science and Technology, Nanjing 210044,*
*China*
**Abstract**
Characteristics of aerosol optical properties, morphologies and their relationship were studied
in urban Beijing during the clear, haze and fog episodes, sampled from 24th May to 22nd Jun,
2012.  Transmission Electron Microscope (TEM), a Cavity Ring Down Spectrometer (CRDS),
a nephelometer and an aethalometer were employed to investigate the corresponding changes
of the aerosol properties. Five episodes were categorised according to the meteorological
conditions, composition and optical variation. Results show the clear episode (EP-2 and EP-4)
featured as the low light extinction with less pollutants, which are mostly externally mixed.
Coarse particles were scarcely observed in EP-2 due to the washout of a previous heavy rain.
Thus the size distribution in EP-2 was smaller than EP-4, which had some mineral particles
introduced from the north. In contrast, industry-induced haze (EP-1) and biomass burning-





induced haze (EP-5) were both impacted by the south air mass. Higher AOD (Aerosol Optical
Depth) values illustrated heavy loading particle concentrations. Due to the collision, size of
most particles was larger with the diameter of 1 μm, resulting in a higher scattering coefficient.
However, as the influence of severe crop residue combustion, a large fraction of soot was
detected, which sticks to the KCl transformed sulphate or nitrate particles. The light absorption
enhancement was contributed by both Black Carbon (BC) acceleration and other light
absorbing substances. Comparatively, soot fog period detected in EP-3 was mostly internally
mixed with sulphates and nitrates, which revealed themselves after electron exposure. The
larger size distribution was likely to be caused by both hygroscopic growth and collision. More
internally mixed particles were observed, which favored the light absorption. The comparison
of all the episodes provides a deeper insight of how mixing states influence the aerosol
extinction properties and also a clue to the air pollution control in the crop burning seasons.
**Keywords:**
Aerosol optical depth, Ångström exponents, Single scattering albedo, Transmission Electron
Microscope, Biomass burning, Soot
**1. Introduction**
Aerosol particles are ubiquitous in the troposphere and exert an important influence on global
climate and the environment (Ramana et al., 2010). They affect climate through direct
scattering, transmission, and absorption of radiation, or indirectly by acting as nuclei for cloud
formation (Buseck and Posfai, 1999). In addition, light extinction by aerosol particles can



impair visibility, both during extreme events such as dust storms, and more widely in the
vicinity of urban regions, frequently leading to regional haze and fog events (Wang et al.,
2009a;Chameides et al., 1999;Sun et al., 2006). Common scattering aerosols in the atmosphere
include inorganic salts and light-color organic carbon. These aerosols have mainly a "cooling
effect" on the climate due to a decrease in the solar radiation that reaches the Earth's surface
(Buseck and POsfai, 1999). Soot aerosols, mineral dust, and brown carbon are important
absorbing aerosols that can lead to global and regional warming effects (Buseck and POsfai,
1999;Bahadur et al., 2012;Wang et al., 2014). The impact of aerosols on the Earth's climate is
a major uncertainty in climate change models as was emphasized in the latest
Intergovernmental Panel on Climate Change (IPCC) report (Solomon, 2007). It follows that
understanding aerosol optical behaviour and associated spatial and temporal variability is a
necessary prerequisite to understanding its role in climate and the environment (Langridge et
al., 2012;Che et al., 2014).
Soot is a major contributor to Earth's radiative balance (Ramana et al., 2010). Recent
investigations involving direct atmospheric measurements of soot aerosols suggest that they
may have a global warming potential second only to $CO_2$, and the warming effect by soot nearly
balances the net cooling effect of other anthropogenic aerosols (Jacobson, 2001). Not
surprisingly, the importance of soot to climate change has been a major focus of many
modelling, laboratory, and field studies (Zhang et al., 2008;Adler et al., 2010;Moffet and
Prather, 2009;Adachi and Buseck, 2008;Ram et al., 2012). The main uncertainty stems from
the fact that the actual amount soot warms our atmosphere strongly depends on the manner and





degree in which it is mixed with other species, a property referred to as mixing state (Jacobson,
2001;Moffet and Prather, 2009). The mixing state was found to affect the soot global direct
forcing by a factor of 2.9. It has been shown that absorption by soot increases when soot
particles are internally mixed and/or coated with other less absorbing materials (Moffet and
Prather, 2009). This enhanced absorption in such structure is because of the lensing effect of
coated materials (Jacobson, 2001). Field measurements indicate that during transport from the
sources, fresh soot becomes internally mixed with sulphate and organics, leading to
enhancement in light absorption, which confirms the modelling calculation (Kleinman et al.,
2007;Doran et al., 2007;Carabali et al., 2012). Kleinman et al. observed a doubling in the ratio
of aerosol light absorption in aged air masses compared to fresh emissions over the eastern U.S.
(Kleinman et al., 2007). Similar increases in absorption by soot-bearing aerosol have been
reported from ground site measurements performed at a series of locations downwind of
Mexico City (Doran et al., 2007). Compiling both the surface and aircraft measurements,
Ramana et al. recommended that the solar–absorption efficiency of the Beijing and Shanghai
plumes was positively correlated with the ratio of soot to sulphate (Ramana et al., 2010). Lei et
al. further confirmed that the enhanced absorption of mixed aerosols depended upon
hygroscopicity and the thickness of the coating (Lei et al., 2014). Based on the combined proof
from the modelling and field studies, most of researchers proposed that internal mixing models
of soot present more realistic absorption estimates as compared to external mixing models in
which soot particles coexist with other particles in a physically separated manner (Jacobson,
2001;Ramana et al., 2010;Lei et al., 2014).




Biomass burning is by far the largest source of primary, fine carbonaceous aerosols in the
atmosphere (Habib et al., 2008). It is estimated to contribute 20% of soot aerosols from biomass
burning. Besides strongly absorbing soot particles, high amounts of brown organic carbon, such
as "tar ball" or HULIS, can be emitted from biomass burning (Roden et al., 2006;Hand et al.,
2005;Hoffer et al., 2006). Brown carbon has a significant absorbing component at short
wavelengths that may be comparable to the soot absorption (Alexander et al., 2008;Bahadur et
al., 2012). Consequently, organic carbon from biomass burning may also contribute to the
warming potential of aerosols (Alexander et al., 2008). These large quantities of climate-related
aerosols can persist in the atmosphere for several weeks and be transported over long distances.
As a result, biomass burning aerosols have a significant impact on climate, which was
considered to provide a major uncertainty in accurately predicting the effects of light-absorbing
aerosols on the climate (Bahadur et al., 2012). Many field measurements in East Asia, South
Asia and Africa have shown extensive biomass burning in these regions causes important
perturbations to Earth's atmosphere (Gustafsson et al., 2009; Alexander et al., 2008; Hand et
al., 2005). Once biomass burning particles are mixed with other atmospheric components
during aging and transport, such as sulfate and dust, solar absorption is further amplified due
to the formation of internally mixed particles (Ramanathan et al., 2005). Such mixtures of
absorbing and scattering aerosols at the regional scale are referred to as ABCs, for atmospheric
brown clouds (Ramanathan et al., 2007). ABCs radiative forcing can cool the surface, stabilize
the atmosphere, and reduce evaporation and monsoonal rainfall. The large influence of ABCs





on the climate and hydrological cycle changes has recently been demonstrated through model
simulations (Ramanathan et al., 2007; Ramanathan et al., 2005).
In the farmlands of eastern China such as that near Beijing, most wheat straw is burned in
the field within one week after harvesting in preparation for rice cultivation during May and
June. Emissions from the biomass burning are often transported and mixed with urban pollution,
leading to degradation of air quality, visibility impairment, and regional haze events (Li et al.,
2010). Stagnation occurs during episodes of urban haze, when there is insufficient wind
velocity to carry pollutants away from the city (Katrinak et al., 1993; Sun et al., 2006). During
these periods of pollutant retention, haze particles aggregate continue to collide and combine,
resulting in larger average sizes and altered morphology (Li et al., 2010). Enhanced absorption
is mainly brought about in the presence of high levels of non-absorbing hygroscopic aerosols
such as sulphates, nitrates, and water-soluble organic carbon, as their hygroscopic nature favors
internal mixing/core-shell formation (Bahadur et al., 2012). On the other hand, under the
condition of high atmospheric relative humidity (RH), the initially hydrophobic soot particles
can become associated with hygroscopic materials, leading to increased scattering due to
particle growth. At an extreme case, the coating material can cause the absorbing fractal soot
to collapse, potentially changing optical behaviour, to further complicate this picture (Zhang et
al., 2008; Langridge et al., 2012; Lei et al., 2014). Such changes cause both positive and
negative effects on the interplay between the direct and indirect aerosol effects, making overall
prediction of the radiative forcing difficult. Up to date, large uncertainties exist in estimates of
the radiative forcing of haze particles because of the lack of detailed in situ measurements of





the mixing state and the associated optical properties as a function of particle size and
composition (Moffet and Prather, 2009). These uncertainties limit our ability to quantify the
relative impacts of soot on climate, thus limiting our ability to make effective policy decisions.
In an attempt to address this knowledge gap, and in the absence of the opportunity for
widespread field studies in eastern China, the experiments in this study were designed to
simultaneously measure mixing states and optical properties of haze particles. The present
analysis focused on the Beijing plume, which in addition to strong urban emissions is
influenced by local agricultural emissions (Li et al., 2010). Light extinction and scattering
coefficient was measure with a cavity ring–down spectrometer (CRDs) and a nephelometer,
respectively. Absorption was calculated from the difference between extinction and scattering.
Individual aerosol particles were identified with transmission electron microscopy (TEM).
Back trajectory analyses suggest flow patterns consistent with long-range transport of
agricultural smoke to the study site during periods when the sampling site was engulfed by the
serious haze and fog.
**2. Experimental Sections.**
**2.1 site description**
All ambient investigation of aerosol optical properties and TEM samplings were conducted
at the Institution of Atmospheric Physics (39°58′N, 116°22′N), Beijing, China, from 24th May
to 22nd Jun, 2012. Samplers were mounted on the roof of a two-story building about 8 m above
ground level. The surroundings are in the convergence of residential and commercial zones
with some steel plants locating around in a distance of 6 to 25 km and a waste incineration





facility (Gaodun) 8 km in the northeast, which has an operational capacity of 1600 t d$^{-1}$. In
addition, the sampling site is suited in the middle of the North Third Ring Road and North
Fourth Ring Road, approximately 360 m south and 380 m north, respectively.    The sampling
site is impacted by the mixture of residential, industrial, waste combustion and vehicle
emissions, but not dominated by any one source.
2.2 **Cavity ring-down spectrometer and nephelometer**
A self-designed cavity ring-down spectrometer (CRDS) is performed to measure the
extinction coefficient of aerosols at 1 min intervals with an accuracy of 0.1 Mm$^{-1}$. Aerosols
were dried by diffusion drying tubes before they reached the CRDS and Nephelometer to
exclude the influence of Relative humidity (RH) on the aerosol optical properties. RH is kept
below 40% to minimize the effects of changing RH on measurements. The cavity is formed by
two high-reflectivity dielectric mirrors (Los Gatos Research, Inc., Mountain View, CA, USA)
and a stainless steel cell equipped with two inlets at both ends and one outlet in the middle. The
entire distance of two mirrors is 76.4 cm, while the filling length is 58.0 cm.    Dry nitrogen is
released near the mirrors at a flow of 0.03 L min$^{-1}$ to prevent the contamination of mirrors and
aerosol flow is set 1.0 L min$^{-1}$. The 532 nm light pulse (energy 100 μJ, duration 11 ns) is
generated by a Q-switched pulsed laser (CrystaLaser QG-532-500). Leaking light through the
mirrors is monitored by a Hamamatsu R928 photomultiplier.    Details about the system were
reported by Li et al (2011). To calculate the decay time, 1000 ring-down traces are averaged at
1000 Hz repetition rate. The extinction coefficient ($\alpha_{ext}$) has an uncertainty below 3% under
the controlled conditions. It is calculated according to the following equation:





$$\alpha_{ext} = \frac{L}{lc}(\frac{1}{\tau} - \frac{1}{\tau_0}) \quad (1)$$

Where L is the length of the cavity, $l$ is effective length occupied by particles, $c$ is the speed of light, $\tau_0$
is ring-time time of the cavity filled with particle-free air and $\tau$ is the calculated decay time (Li et al.,

170    2011).

An integrating nephelometer (TSI, Model 3563) is operated to obtain aerosol scattering coefficient at three
different wavelengths (450, 550, and 700 nm) and the flow rate is set at 5 L min$^{-1}$. During the field campaign,
zero check is done automatically by pumping in particle-free air for 5 min once every 2 h, and a span check
is conducted manually using $CO_2$ as the high span gas and filtered air as the low span gas every week. RH
is kept below 40% to minimize the effects of changing RH on measurements (Peppler et al., 2000;Clarke
et al., 2007). The raw data were corrected for truncation errors and a non-Lambertian light source using
Ångström exponents (å) according to Anderson and Ogren (1998) (Anderson and Ogren, 1998). Generally,
the total uncertainty of the scattering coefficient ($\alpha_{scat}$) is generally below 10%. In accordance with   the
extinction coefficient at 532 nm, the scattering coefficients is converted to 532 nm
($\alpha_{scat,532}$) on the basis of the following equation:
$$\alpha_{scat,532} = \alpha_{scat,\lambda}(\frac{532}{\lambda})^{-\overset{\circ}{a}} \quad (2)$$

Where $\alpha_{scat,\lambda}$ is the scattering coefficient at the wavelength of λ. Accordingly, å can be computed
calculated as the equation (3),
$$\overset{\circ}{a} = -\frac{\lg(\alpha_{scat,\lambda_1} / \alpha_{scat,\lambda_2})}{\lg(\lambda_1 / \lambda_2)} \quad (3)$$

and the single scattering albedo ($\omega_0$) at the given wavelength can be calculated from equation (4),
$$\omega_0 = \frac{\alpha_{scat}}{\alpha_{ext}} \quad (4)$$





As the sum of absorption ($\alpha_{abs}$) and scattering ($\alpha_{scat}$) coefficients equals the extinction coefficient ($\alpha_{ext}$),
$\alpha_{abs}$ can be derived from the equation (5),
$$\alpha_{abs} = \alpha_{ext} - \alpha_{scat} \quad (5)$$

It is known that RH also has a profound impact on visibility (Chow et al., 2002), however, in this study the
aerosols passed through a diffusion drying tube before the measurement of optical properties, thus    aerosol
optical property measurements and TEM observations were both performed in dry condition.
**2.3 Aethalometer**
An Aethalometer (model AE-31, Magee Scientific Company) was employed to simultaneously quantify
the black carbon (BC) concentration by calculating the optical attenuation (absorbance) of light from Light
Emitting Diode lamps emitting at seven different wavelengths (370, 470, 520, 590, 660, 880 and 950 nm)
every 5 minutes, with a typical half-width of 0.02 μm (Hansen, 2003). The flow rate was set to be 5 L min⁻
¹ and a clean filter canister in the inlet was used weekly to conduct the zero calibration. A $PM_{2.5}$ cyclone
(BGI SCC 1.828) was employed in the sampling line with a flow rate of 5 L $min^{-1}$. A typical noise level is
less than 0.1 µg $cm^{-3}$ on a 5-min basis. Two photo-detectors monitor the light intensity as a function of time.
One measures the light intensity of the light crossing reference quartz filter, while the other measures that
of the same light crossing a sample spot under the identical conditions. The wavelength at 880 nm was used
to derive the aerosol absorption coefficient ($\sigma_{abs}$). Then BC concentration can be converted under the
assumption that the BC mass concentration [BC] on the filter is linearly correlated to the aerosol absorption
coefficient, as the following equation (6),
$$\sigma_{abs} = \alpha[BC] \quad (6)$$

Where [BC] is the BC mass concentration, and α is a conversion factor. A factor of 8.28 $m^2$ $g^{-1}$ was
employed to convert the aerosol absorption coefficient to BC concentration, according to the results of
inter-comparison experiment conducted in south China (Wu et al., 2009;Yan et al., 2008).
The uncertainty of measurement might originate from the multiple scattering in the filter fibres in the
unloaded filter and in those particles embedded in the filters (Clarke et al., 2007;Jeong et al., 2004). The



attenuation values were within the limit of an acceptable uncertainty, that is, no greater than 150 in the
range of 75-125 at various wavelengths, verifying the reliability of the measurement. Moreover, the BC
concentration was compared with the results of multi-angle absorption photometry (MAAP, Model-5012)
and a particle soot absorption photometer (PSAP, Radiance Research), which shows great consistence.
**2.4 Aerosol optical depth**
Aerosol optical depth (AOD) data at the sampling site is based on the MODIS (Moderate Resolution
Imaging Spectroradiometer) retrieved data from a CIMEL CE-318 sunphotometer (AERONET/PHOTONS)
at Institute of Atmospheric Physics, reflecting the amount of direct sunlight prevented from reaching the
ground by aerosol particles by measuring the extinction of the solar beam. The AOD value of the sampling
site is downloaded from the AERONET (http://aeronet.gsfc.nasa.gov), using the Level 2.0. Quality Assured
Data. These data are pre and post field calibrated, automatically cloud cleared and manually inspected.
The regional distribution of AOD was obtained from Giovanni (GES-DISC Interactive Online Visualization
And Analysis Infrastructure) maps from MODIS satellite data (http://disc.sci.gsfc.nasa.gov/giovanni). Two
continuous episodes featuring as clear and haze are chosen, 23th May to 27th May and 19th June to 27th
June, respectively.
**2.5 TEM Analysis**
Samples were made by collecting air-borne particles onto copper TEM grids coated with carbon film
(carbon type-B, 300-mesh copper, Tianld Co., China) using a single-stage cascade impactor with a 0.5 mm
diameter jet nozzle at a flow rate of $1.0 \, L \cdot min^{-1}$. According to the visibility, the sampling time varies from
1 min to 10 min. 3 or 4 samples were collected each morning at around 8 am and also each time when haze
or fog appeared. After collection, samples were stored in a dry plastic box sealed in a plastic bag and kept
in a desiccator at 25 ℃ and 20 ± 3%. Details of the analysed samples, such as sampling time and
instantaneous meteorological state are listed in Table 1.
Individual aerosol samples were analysed using a high resolution TEM (JEOL 2010, Japan) operated at
200 kV. The TEM can obtain the morphology, size, and mixing state of individual aerosol particles. Energy-
dispersive X-ray spectrometer (EDS) can get the chemical compositions of the targeted particles. Cu and C



were excluded from the copper TEM grid with carbon film. Details can be found in our previous paper (Fu
et al., 2012; Guo et al., 2014a). Since particle sizes on the grid decrease from the centre to the periphery
due to the limitation of sampler, to ensure the representative of the entire size range, three to four round
meshes were chosen from the centre to the periphery in a line. Each mesh analyses three to four views. The
average values of each mesh were used for statistics. The analysis was done by labour-intensive manual
sortation of the particles. 9 grids, all together of 1173 particles have been analyzed by the TEM.
**2.6 Back trajectories and meteorological data**
NOAA/ARL Hybrid Single-Particle Lagrangian Integrated Trajectory model (available at
http://www.arl.noaa.gov/ready/hysplit4.html) was employed to determine back trajectories arriving at
Beijing at 100 m, employing the data of global data assimilation system (GDAS). Each trajectory
represented the past 72 h of the air mass, with its arrival time at 00:00 UTC every day.
Meteorological data was downloaded from Weather Underground (www.wunderground.com), and Daily
$PM_{10}$ values were transformed from daily API (Air Pollutant Index) in the datacentre of ministry of
Environmental Protection of the People's Republic of China (http://datacenter.mep.gov.cn/).
**3. Results and Discussion**
**3.1 Episodes segregations**
Haze is usually defined as a weather phenomenon that lasts a duration of at least 4 h when the visibility is
less than 10 km and RH lower than 80% (Sun et al., 2006), while fog is characterized with a higher RH,
larger than 90% in contrast, according to the Chinese Meteorological Administration. The sampling period
was categorized into 5 episodes to study the optical properties between different weather phenomena (Fig.
1). Although every episode contains a mixture of different pollutions, the main origin can be discerned by
studying the weather condition, back trajectories and fire maps. The 1st episode (EP-1) was from 28th May
to 29th May, when a haze occurred with the south wind bringing in the industrial pollution from the heavily
polluted cities in the south. This conformed to the 3-day back trajectories in Fig. 2a, showing the air masses
passing through Henan, Shandong, Hebei and Tianjin before arriving at the sampling site. Only scattered
fire spots were observed during these days along the air mass pathway, suggesting little biomass burning



emission interference. The 2nd episode (EP-2) was in clear weather on 30th May. A heavy rain interrupted
the previous haze; hence residuals were cleaned up by rain washout effect. It was impacted by the air mass
from the north region (Fig. 2b), as the north wind was relatively clean and the time was insufficient for a
heavy accumulation, this episode can be viewed as the background. The 3rd episode (EP-3) from 31st May
to 9th Jun was fickle, with a variety of transitions between fog, haze and clear days. This was partly caused
by the variable wind directions and air mass transferring (Fig. 2c). When the wind is from east side and the
back trajectories are across the Bohai Sea, the air mass carries a high content of water vapour, facilitating
the formation of fog, whereas when south wind is dominant, haze is likely to occur (Wang and Chen,
2014;Zhang et al., 2010). The following 4th episode (EP-4) from 10th Jun to 16th Jun was mainly clear days
with slight dust. Their back trajectories originated from the north part (Fig. 2d), mostly travelling from the
Siberian region, across eastern Mongolia and Inner Mongolia and arriving the site with little pollution but
a few dusts. The last episode (EP-5) was from 17th Jun to 21st Jun. Severe haze was observed in a long
duration. Together with the fire spots, Fig. 2e shows that the air parcel pathway across by dense fire spots,
indicating a severe impact of the biomass burning. Every year after harvest, crop residue burning is
extremely frequent in Anhui, Shandong and Henan provinces as they served as important centres for the
rice supply (Li et al., 2010)). Therefore, the biomass burning emissions can be the main contributor to the
haze formation in this episode. Besides the crop combustion, the high-loading fine metal particles observed
imply the influence of industrial sources.
**3.2 Optical parameter variation**
Aerosol optical depth (AOD) is representative of the airborne aerosol loading in the atmospheric column,
which is also verified by a significant related coefficient with $PM_{10}$ (R=0.603) (Fig. 1). The overall AOD
is contributed by both Mie scatter and Rayleigh scatter (Fig. 3a). The former one is produced by the scatter
effect of particles while the latter one by gases. Fig. 1 demonstrates that gas plays a negligible role in the
AOD value, especially when aerosol loading is high. Apparently, the AOD value varied with the weather
transition. During the clean days, the mean AOD was 0.723, while it became higher when the haze and fog





were formed, with a mean value of 2.92 and 1.14, respectively. Apart from the rain, AOD reached its highest
value of 5.0 in the hazy EP-5, which was more than 5 times than the average AOD of 0.95 in Beijing
measured from Mar 2012 to Feb 2013 (Guo et al., 2014b). Such high AOD could be attributed to the
pollutant accumulation, especially biomass burning emission from the crop combustion.

293        Ångström exponent (å) is a good indicator of aerosol size distribution, which decreases with the increase

of particle size (Eck et al., 1999). The value is computed from pairs of AOD measurements at 700 nm with
450 nm, 700 nm with 550 nm and 550 nm with 450 nm, respectively. A high accordance is observed
between each pair (Fig. 3b). The å increases sharply to its highest value above 2.0 at EP-2, 45 times of the
minimum value 0.044 observed in EP-5. This could be explained by the wet removal impact of the heavy
rain. It is well known that rains wash out the coarse particles, resulting in a fine size distribution (Dey et al.,
2004). The å value during EP-4 fluctuated between 0.08 and 0.2. Since the rains are light and short, the
clear days in EP-4 are more impacted by the north air mass, which brings in a larger fraction of coarse dust
particles. Comparatively, the å value was lower in both the haze and fog period including EP-1, EP-3 and
EP-5. Especially in the case of EP-5, the low å value indicated that the biomass burning emission could
contain more coarse particles. Such scene is in contrast to the conclusion that the haze days were dominated
by fine particles (Yan et al., 2008). It is likely caused by the high collision occurrences of fine particles
along the long-range transport from the fire spots (Wang et al., 2009b). In comparison, the å value during
2001 to 2005 in Beijing altered between 0.04 and 1.06 (Yu et al., 2006). The lower limit is similar with the
present field-measurement, while the upper limit is much higher than this study. This could be attributed to
the increase of fine particle emission contributed by more vehicles, waste incineration and industrial plants
during these years.

310        Single scattering albedo (SSA), $\omega$, is defined as the ratio of the aerosol scattering coefficient ($\sigma_{sca}$) to the

extinction coefficient ($\sigma_{ext}$). This parameter is especially important in the estimation of direct aerosol
radiative forcing, since even a small error in its estimation might change the sign of aerosol radiative forcing
(Takemura et al., 2002). Figs. 3c and d show the time series of $\sigma_{sca}$, $\sigma_{abs}$, $\sigma_{ext}$ and SSA at 550 nm during

314 the sampling period. The mean ω was 0.73, 0.82 and 0.79 in EP-2, EP-4, and EP-5, respectively, implying

315 that mineral dust in EP-4 accelerates the optical scattering while soot favours the optical absorption.

316 Compared with other reported results (Che et al., 2014; Li et al., 2007; Qian et al., 2007), the mean ω is

317 lower in this study, which supposed that more soot is presented. Research shows soot emission is much

318 higher in recent year, mainly contributed by the residential coal combustion, also biomass burning, coke

319 production, diesel vehicles and brick kilns (Wang et al., 2012b). Especially when air masses moved from

320 south direction to the sampling site aerosols were influenced by heavy soot-sulfate-OC-mixed pollution

321 from the dense population centres and industrial areas and substantial secondary aerosols are produced

322 during transport (Sun et al., 2006;Wang et al., 2006), which was also confirmed in the TEM observation.

323 In these conditions, SSA was even lower than those of the regions along the pathway.

324 **3.3 TEM analysis**

325 Based on morphology and chemical composition, we classified over 1173 particles into 9 categories: S-rich

326 (Fig. 4a), N-rich (Fig. 4b), mineral (Fig. 4c), K-rich (Fig. 4d), soot (Fig. 4e), tar ball (Fig.4f), organic

327 (Fig.4g), metal (Fig. 4h) and fly ash (Fig. 4i). The classification is similar to that adopted by Li Weijun (Li

328 and Shao, 2009).

329 The most common particles are sulphates and nitrates (Figs. 4a, b), which are of the size around 1.0

330 μm, and have a light scattering ability when externally mixed (Jacobson, 2001). Sulphates appeared as

331 subrounded masses under the TEM. They decomposed or evaporated under the electron beam exposure.

332 Conventionally, they were formed by the reaction of precursor $SO_2$ or $H_2SO_4$ with other gases or particles

333 (Khoder, 2002). Nitrates were mostly of scalloped morphology in the TEM view. They were relatively

334 stable when exposed to the electron beam. Nitrates formed through either the homogeneous reaction with

335 the precursor $NO_2$ or heterogenic reaction with $HNO_3$ (Khoder, 2002), but they possess a slower reaction

336 rate than sulphates. Thus, they usually follow the sulphate formation (Pathak et al., 2004; Seinfeld and

337 Pandis, 2012).

338 In the clear episodes influenced by the northern air mass, dust particles are relatively abundant. Dust

339 particles (Fig. 4c) were large in size, usually bigger than 1.0 μm, up to 8.0 μm under our sampling conditions.



Their compositions differed from each other, mostly are silicates and calcium sulphate or carbonate, which
were all stable to the electron beam. Dust particles were reported to have a light scattering effect, thus
resulting in negative aerosol radiative forcing (Wang et al., 2009b). They had a large portion in EP-4,
impacted by the north wind from the dusty regions.
As to the biomass burning related haze, notable tracer particles are presented, such as K-rich particles
(Li et al., 2010;Duan et al., 2004;Engling et al., 2009), soot (Li et al., 2010), tar ball (Chakrabarty et al.,
2010;Bond, 2001) and organic (Lack et al., 2012). K-rich particles (Fig. 4d) often exist as sulphate or nitrate.
A larger fraction of K-rich particles was found in the EP-5 than other periods. Together with the back
trajectories and fire spot maps, it further confirmed that the regional haze occurred in EP-5 was contributed
significantly by the field combustion. K-rich particles were characterized by the irregular shape which is
unstable when exposed to electron beam. KCl was barely detected in the samples, even though it has been
found to be internally mixed with $K_2SO_4$ and $KNO_3$ in fresh biomass burning plumes (Li et al., 2010;Li et
al., 2003;Adachi and Buseck, 2008). K-rich particles in our measurements showed that mostly they consist
of N, Na, O, S, and K and are free of Cl, implying KCl could have suffered    from chemical reactions and
changed into sulphates or nitrates (Li and Shao, 2010), thus displaying negative climate forcing under
externally mixed conditions.
Soot (Fig. 4e) is vital to light absorption, which can alter regional atmospheric stability and vertical
motions, as well as influence the large scale circulation and precipitation with significant regional climate
effects (Ramanathan et al., 2001;Jacobson, 2002). It is another indicator of biomass combustion with a
structure like onion ring, resembling a fractal long chain as agglomerates of small spherical monomers (Li
and Shao, 2009). The fresh soot was loose and externally mixed. However, after undergoing a long-range
transportation, it has become more compacted, with a slight increase of O concentration because of the
photochemistry (Stanmore et al., 2001). Meanwhile, soot generally attaches to other particles on the surface
or serves as the core for other particle formation as the result of its stable and insoluble nature.
Tar ball (Fig. 4f) is a spherical carbon ball with a small fraction of O. It was thought to origin from the
smouldering combustion and have strong absorption effects (Chakrabarty et al., 2010;Bond, 2001). Tar





balls constituted a large fraction of the fresh emitted wildfire carbonaceous particles (China et al.,
2013;Lack et al., 2012). But it was seldom found in our study, even in EP-5 when there was severe biomass
burning emission. This may be caused by the difference in burning species. Most are wheat and rice residues
in this study, which have relatively less tar ball emission. An indoor chamber combustion simulation of the
different crop residues was conducted in our lab and confirms the speculation.
Organic matter (Fig. 4g) is amorphous specie and have a low contrast under the TEM view, which is
stable with strong electron beam exposure. It can be traced to the direct emission such as biomass burning
(Lack et al., 2012), or the second reaction between the VOCs with ozone (Wang et al., 2012a). It can absorb
radiation in the low-visible and UV wavelengths (Chakrabarty et al., 2010;Clarke et al., 2007;Lewis et al.,
2008;Hoffer et al., 2006). In addition, when compassing soot as the core, organic matter can enhance
absorption by internal mixing (Adachi and Buseck, 2008).
For the common haze and fog episodes, the stagnated weather favors the accumulation of pollutants,
especially metal particles and fly ash. Metal particles (Fig. 4h) are generally round and stable under the
TEM view. Less optical properties are discussed about the metal particles. Details of the classification of
metal species and origins were reported by Hu et al. (Hu et al., 2015). Fly ash (Fig. 4i) was a dark sphere
with the diameter usually below 1 μm. It is a common product of industrial activities in the northern China
(Shi et al., 2003). As the complex refractive index (CRI) indicates, metal oxide particles and fly ash can
scatter light, but the former has a weak absorption ability while the later has almost no light absorption
ability (Ebert et al., 2004).
Figure 5 shows percentage of the 9 components in clear, haze and fog episodes under external, internal
and adjacent states (partially internal). About 28% of particles were internally mixed in the foggy days,
while about 52% of particles exhibited external mixing state in clear days through TEM analysis. Haze and
fog episodes had a higher possibility of collision due to the heavy particle loading, leading to a higher
adjacent state percentage around 65%.
**3.4 Optical properties related to morphological of aerosols**





The differences of the particle morphologies under different weather conditions can be easily observed,
and the results are shown in Fig. 6. Due to the washout effect of the heavy rain, particles in the typical clear
period EP-2 were normally smaller in size under the TEM (Figs. 6a, b), verifying the larger Ångström
exponent. The coarse particles, such as dusts, were hardly observed, whereas a few K-rich particles were
detected, of which presented in small cubic shape. Such particles could be attributed to the near-by coal
combustion around the sampling site due to the slight fire spots presence. Besides, the cubic shape of K-
rich particles suggested they have not undergone long transportation or severe photochemical reaction.
Likewise, soot was generally less oxidized in the haze and fog periods, maintaining fractional and externally
mixed. As the result of the existence of uncovered and less soot, SSA was higher. Small metal particles and
amorphous Zn-particles dominated the fine particles, which were attributed to the industrial activity and
waste incineration (Choël et al., 2006;Moffet et al., 2008).
In the biomass burning induced EP-5, the increase in aerosol loading played a remarkable role in the
enhancement of scattering coefficient and decrease of visibility (Kang et al., 2013;Charlson et al.,
1987;Deng et al., 2008). Because of the high rate of aerosol collision, particles were larger than the clear
days under the TEM (Figs. 6c, d), leading to a smaller Ångström exponent. Nearly all the soot studied in
this condition was compact and adhesive. It is predominantly adjacent to K-rich particles, which were larger,
rounder or with an outlayer of high S concentration. Together with the soot association on the surface, they
were probably transported from the south crop residual burning places and undergo some chemical changes,
verifying the trajectories passing through intense fire spots. Due to the high concentration of soot, these
haze periods are characterized with a high absorption coefficient, which was in accordance with the results
displayed in the Figure 3.
The BC variation during the sampling period was illustrated in Fig. 7. The preliminary component of BC
can be viewed as the soot.   High BC concentration was easily recognized in EP-5 with a mean value of
12.8 μg m$^{-3}$, while it is 1.04 μg/m$^3$ during clear periods. The former was about 11.3 times higher than the
later. In comparison, absorption coefficient of the EP-5 (468.7 Mm$^{-1}$) was about 94.7 times of the EP-4 (1.3



Mm$^{-1}$), more than 8 times of the BC ratio. Models estimated an enhancement of BC forcing up to a factor
of 2.9 when BC is internally mixed with other aerosols, compared with externally mixed scenarios
(Jacobson, 2001), which was much lower than this case. Accordingly, other light absorbing substances may
contribute to the discrepancy. Brown carbon is an indispensable component of biomass burning, which has
a strong absorption ability as well (Hoffer et al., 2006;Andreae and Gelencsér, 2006). Other particles like
dust may also contribute to the over-enhanced absorption coefficient (Yang et al., 2009). Our observations
were accordant with that of radiative forcing from aerosols in regional hazes over northern China, which
shows that aerosol particles under hazy weather conditions generate a positive heating effect on the
atmospheric column (Wang et al., 2009b;Xia et al., 2006).
In foggy days, particles were also generally larger than the clear days in the TEM views (Figs. 6e, f),
resulted from the hydroscopic growth under the high relative humidity, as well as the collision among
overloading particles, which was likewise illustrated by the Ångström exponent. Consequently, the larger
particles enhance the scattering of sunlight, and lead to higher reduction of visibility (Quan et al., 2011).
Moreover, Chow et al., reported that the relative humidity (RH) also has a profound impact on visibility
(Chow et al., 2002a). Some fan-like nitrate particles have inclusions which may act as the growth cores or
be encompassed during the hydroscopic growth. Bian et al. (2009) reported that whenever the RH is
elevated, its importance to the AOD is substantially amplified if the particles are hygroscopic (Bian et al.,
2009). Li et al. (2010) found that soot particles became hydrophilic when they were coated with the water-
soluble compounds such as    $(NH_4)_2SO_4$, $NH_4HSO_4$, $KNO_3$, $K_2SO_4$, or oxidized organic matter, implying
that soot can provide important nuclei for the development of fine particles (Li et al., 2010). Furthermore,
Fig. 6e and 6f also illustrate a large fraction of internally mixed soot. It was not visible until being    exposed
to electron beam for about 30 s. Sulphate and nitrate coatings function like a "focusing mirror", thus light
absorption ability of the internally mixed soot particles are enhanced by 30% than soot alone (Fuller et al.,
1999). However, the BC concentration in foggy conditions was 6.12 μg m$^3$, and the absorption coefficient
is 143.7 Mm$^{-1}$, which were 2.09 and 0.83 times of the hazy days, respectively. The enhancement was much
greater than 30%.    The reason is still not very clear and further investigation is thus needed. A variety of




metal particles were also observed in the foggy days, as foggy days had a stable low upper layer boundary
and slight wind, leading to the accumulation of pollutions. These pollution sources range from steel plants
and waste incineration to vehicle emission and so on (Hu et al., 2015).
**4 Conclusions**
Using the TEM and optical instruments, we investigated the particle morphological impact to the optical
properties by comparing the discrepancies during clear, haze and fog episodes in the summer of Beijing
unban. The clear episodes are most characterised by air mass from the north, which brings in slight mineral
particles. When a heavy rain occurred previously, it can washout the large particles and lower the particle
size distribution. Particle concentration was lower because most of particles existed in the form of external
mixture, leading to the attenuation of light absorption and scattering. Comparatively, haze and fog episodes
have higher particle concentration, larger size distribution and lower visibility. Pollutants such as metal
particles and fly ashes from the industries accumulate under the stagnated weather. In the later June, air
parcels from the intensive crop residue combustion region in the south introduce the biomass burning
emission, such as soot and K-rich particles, resulting in a period of severe haze. Light scattering and
absorption increase notably contributed by the mixed BC and other light absorption substances. In fog
periods, a large proportion of internally mixed particles, especially soot with the coating of nitrates or
sulphates, which favors the light absorption due to the "focusing mirror" effect. With the help of TEM, we
can directly investigate the size, mixing state, and chemical composition of the particles, thus provide
reference to the theoretical calculation of the BC model and ensure more precise interpretations of the
optical properties. However, the TEM analysis is done in the vacuum condition, which may cause the
decomposition and volatilization of some sensitive particles. Besides, the detail transition mechanism
among clear, haze and fog episodes needs higher time-resolved instrument and more statistics. All these
factors should be considered in the future studies.
**ACKNOWLEDGMENTS**



This work was supported by National Natural Science Foundation of China (Nos. 21577022, 21190053,
40975074), Ministry of Science and Technology of China (2016YFC0202700), and International
cooperation project of Shanghai municipal government (15520711200).

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





**Table 1** Details about the analysed samples on the sampling time, instantaneous meteorological state and
the number of particle analysed in each sample.

| Sampling Time (BST[a]) | | | Conditions | RH (%) | Temp. (℃) | Wind | | Visibility (km) | No. |
|---|---|---|---|---|---|---|---|---|---|
| Date | Starting | Duration | | | | Speed (m/s) | Direction | | |
| 25-05-2012 | 13:40 | 4 min | Clear | 20 | 29 | 2 | 160 | - | 136 |
| 30-05-2012 | 9:31 | 16 min | Clear | 29 | 24 | 7 | 350 | - | 92 |
| 02-06-2012 | 9:00 | 1 min | Mist[b] | 83 | 20 | 4 | 180 | 2 | 146 |
| 02-06-2012 | 13:27 | 2 min | Clear | 48 | 27 | 4 | 190 | - | 138 |
| 03-06-2012 | 10:13 | 15 s | Fog | 88 | 22 | 1 | variable | 1.2 | 110 |
| 18-06-2012 | 18:52 | 2 min | Haze | 55 | 29 | 3 | 140 | 3 | 172 |
| 19-06-2012 | 9:10 | 2 min | Haze | 61 | 25 | 1 | variable | 2.8 | 120 |
| 21-06-2012 | 9:10 | 1 min | Haze | 69 | 26 | 2 | 110 | 2.2 | 117 |
| 23-06-2012 | 12:45 | 2 min | Mist[b] | 84 | 25 | 4 | 120 | 3 | 142 |

[a]Beijing standard time (8 h prior to GMT).

[b]Mist is studied here as fog.


703                                                                    32



**Figure captions**

**Figure 1**. 5 episodes categorization. EP-1features haze induced mainly from transportation of south industrial pollution, EP-2 clear, EP-3 frequent transition among haze, fog and clear conditions, EP-4 clear with rain interrupted, and EP-5 haze resulted mainly from the biomass burning.

**Figure 2.** The 3-day back-trajectory clusters of each episode, arriving at Beijing at the height of 100 m, together with the fire spot distribution of these periods.

**Figure 3.** TEM typical views of the particles in clear (upper panel), haze (middle panel) and fog episodes (bottom panel). 9 components are marked with the colourful arrows. (a1) (b1) (c1) (d1) (e1) (f1) is obtained before the electron exposure and (a2) (b2) (c2) (d2) (e2) (f2) is after exposure. A fraction of S-rich particles and other unstable particles decompose after electron exposure.

**Figure 4.** 9 categories of particles under the TEM view. The inserted spectra are obtained by the EDS, and the grid like images are acquired from the SAED. (a) S-rich, (b)N-rich, (c)mineral. (d)K-rich, (e)soot, (f)tar ball, (g)organic, (h)metal, (i)fly ash.

**Figure 5.** Variation of optical parameters during the study period. (a) Total Aerosol optical depth (AOD), and AOD resulted from Mie scatter and Rayleigh scatter; (b) Ångström exponent (å) computed from the pairs of 700 nm and 450 nm, 700 nm and 550 nm, and 550 nm and 450 nm; (c) light extinction, absorption and scattering coefficients; (d) calculated single scattering albedo (SSA).

**Figure 6.** Percentages of 9 particle components under clear, haze and fog conditions with different mixing states.

**Figure 7.** BC concentrations converted from the data measured by AE-31 and MAAP. Good correlation is observed.






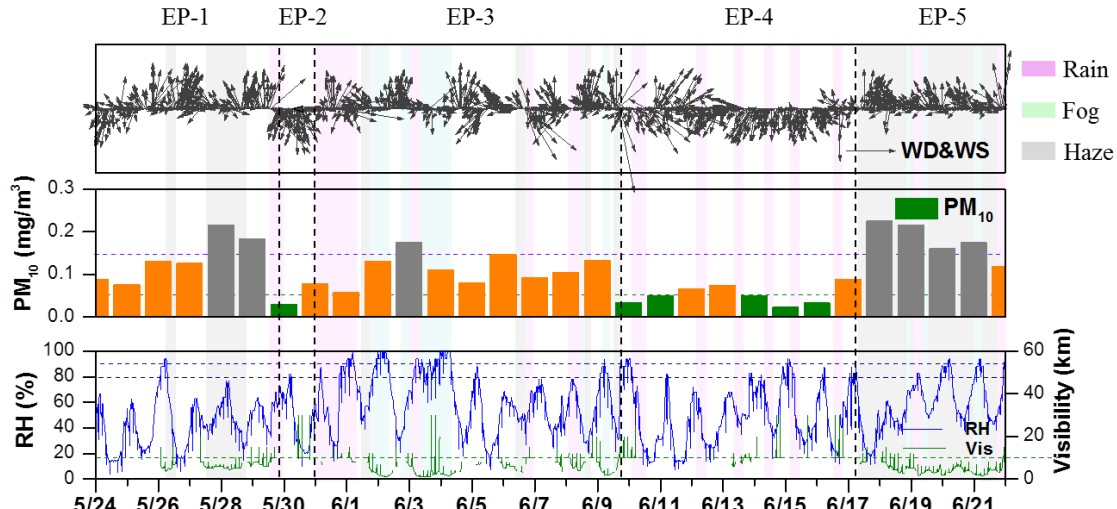











737                                                                 **Figure 1**






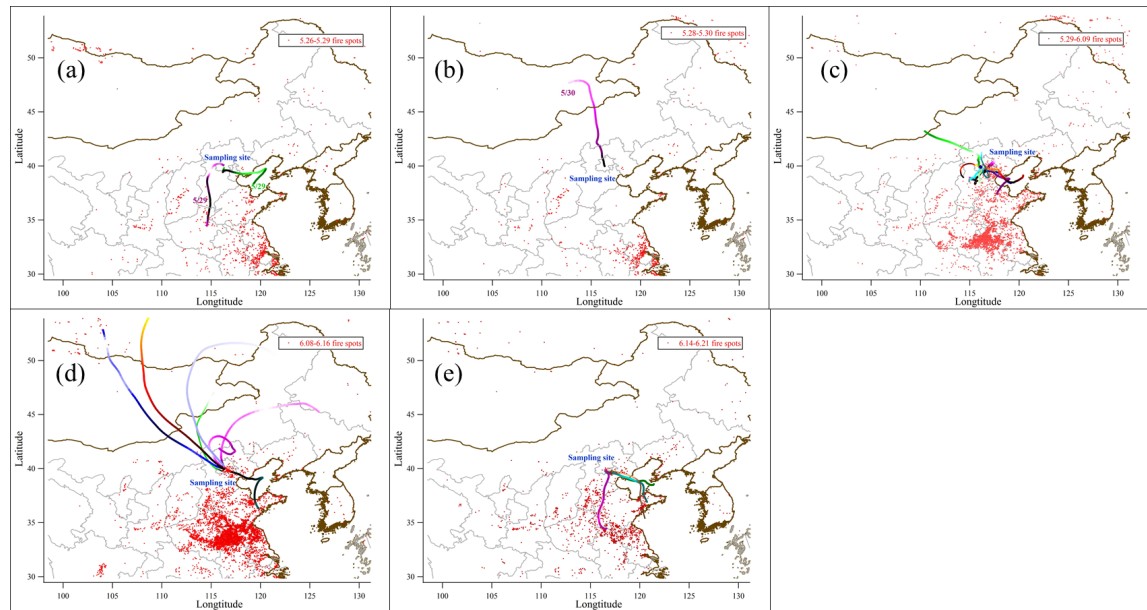







**Figure 2**





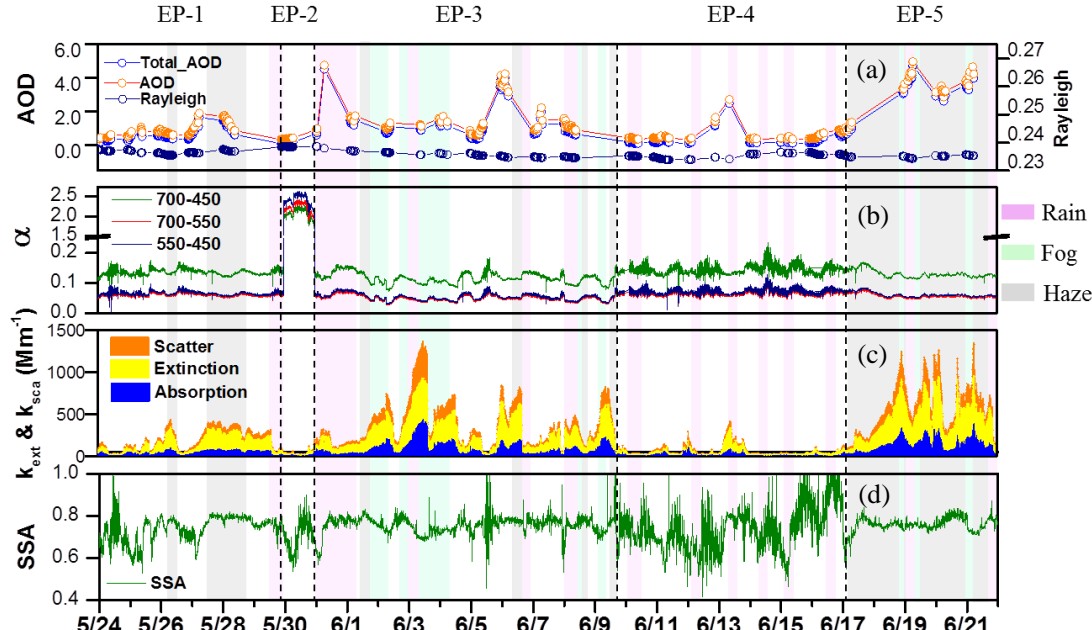







**Figure 3**











761                                                           **Figure 4**






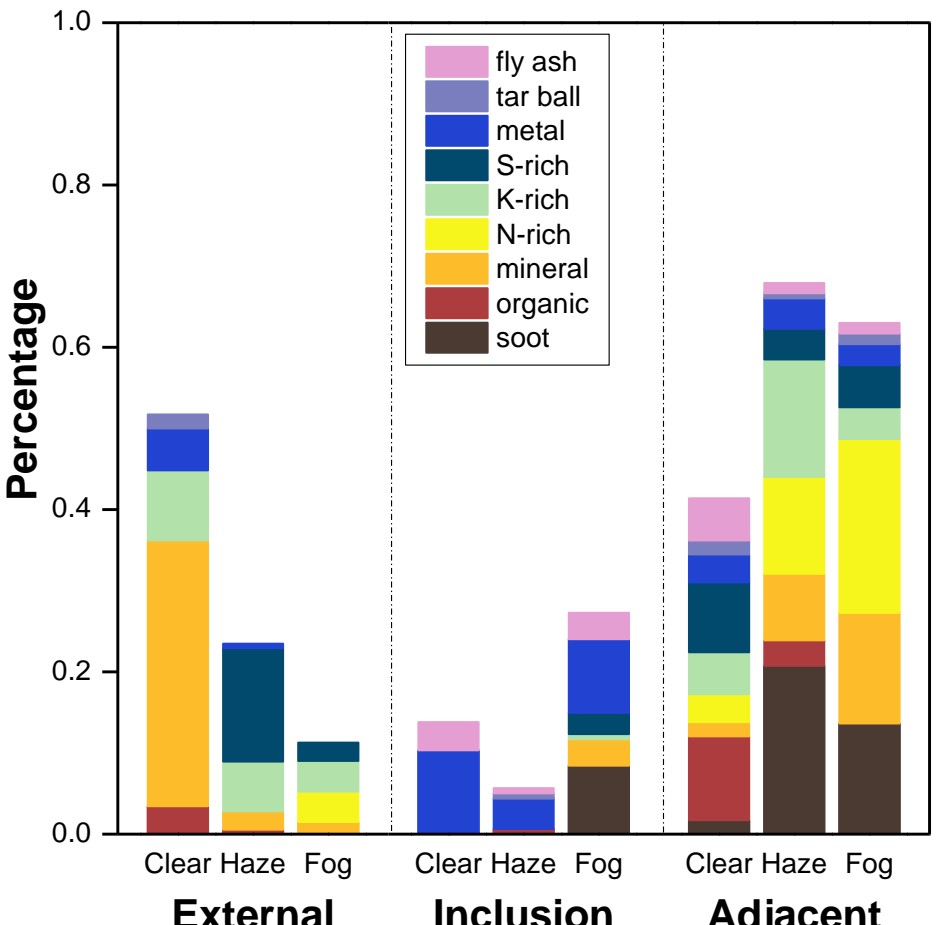




766                                                                        **Figure 5**





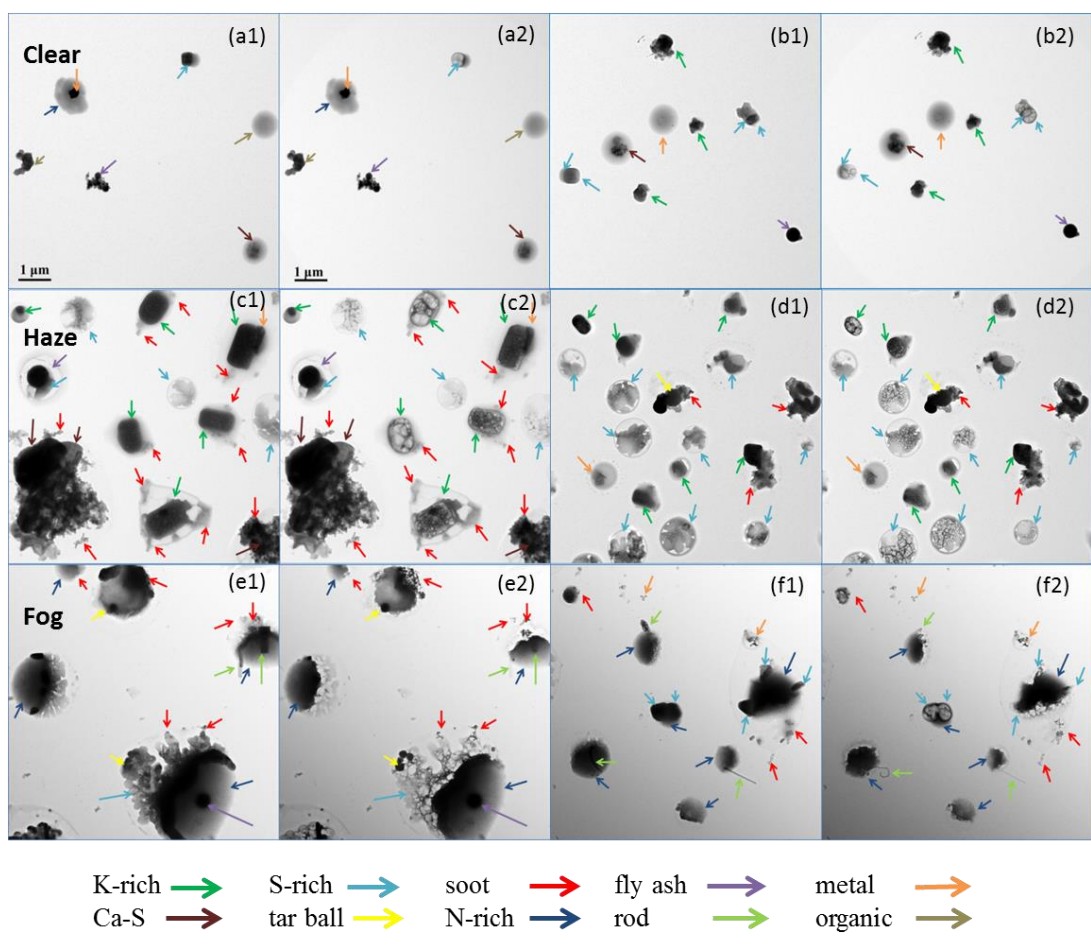






772                                                                        **Figure 6**





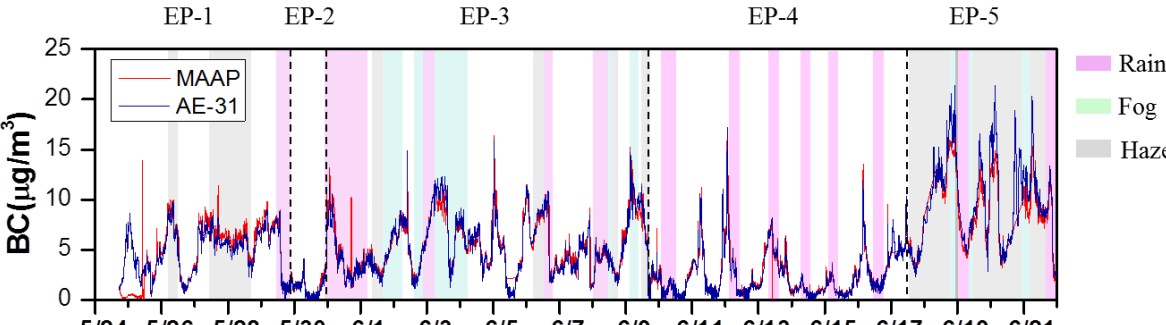










**Figure 7**