# Peer review of "Real-Time Aerosol Optical Properties, Morphology and Mixing States under Clear, Haze and Fog Episodes in the Summer of Urban Beijing"

_Atmospheric Chemistry and Physics, 2016_

## Referee Comment (RC1) · Anonymous Referee #1 · 23 Jan 2017

The authors investigated the optical properties, morphologies and their relationship of aerosol particles collected in a typical haze pollution in the atmosphere of urban Beijing, China. The optical properties of aerosols played an important role in climate change. Up to date, little information about the influence of aerosol morphologies on the optical properties of haze particle is available. So, the topic focused by this manuscript is important and the authors presented interesting results. The result presented herein could make some contribution to the field of atmosphere chemistry. However, the technical quality of the manuscript is not ready for publication. Especially, English writing should be improved greatly. The special comments are as follows:

1. The English thorough the manuscript are suggested to be improved by an English

native speaker. 2. The abstract immediately begins with specific results, without providing an overview of the study or its objectives. 3. Line 28: Please give a definition of soot fog period. 4. Line 284: the abbreviate of correlation coefficient is "r" not "R". The valid number thorough the manuscript should be consistent. 5. Line 371: Is "organic matter" a right expression? Tar ball and soot are also organic matter. The statement that organic matter could be traced to the direct emission such as biomass burning or the second reaction between the VOCs with ozone is speculative. More discussion should be conducted to demonstrate this statement. 6. Line 396: How does the cubic shape of K-rich particles suggested they have not undergone long transportation? More clarification needed. 7. TEM analysis can only observe limited amount of particles. Could you assure the quality of the data? 8. More literature published after 2016 about the optical properties should be cited.

Please also note the supplement to this comment:
http://www.atmos-chem-phys-discuss.net/acp-2016-976/acp-2016-976-RC1-supplement.pdf

---

## Referee Comment (RC2) · Anonymous Referee #2 · 16 Feb 2017

General comment: This manuscript investigated the aerosol optical properties and morphologies in a typical urban site of Beijing for one month during summer of 2012. The investigation combined the optical properties with the morphologies provided very useful information about the aging effects of aerosol on their optical properties, confirming that the internally mixed particles formed by aging biomass emission during the wheat harvesting period in the region greatly increased the absorption coefficient. However, the presentation of the manuscript is suggested to be improved before publication. Specifics: 1.The English thorough the whole manuscript should be improved by a native English speaker. 2.More recent reports about the influence of agricultural activities including biomass burning on the regional air quality are encouraged to be

cited. 3.Abstract: Line 12-16, the sentences are suggested to be replaced by "Aerosol optical properties and morphologies were measured by TEM, CRDS, a nephelometer and an aethalometer in a urban site of Beijing from 24 May to 22 June". The clear, haze and fog episodes just occurred during the sampling period, it didn't need to mention them in the abstract. The phrase of "sampled from..." is not correct in English grammar. The instruments were used for measuring the aerosol properties, but not for investigating the corresponding changes of the aerosol properties. Line 16- 17, the sentence is meaningless, because the individual episode was mentioned in the following sentences. Line 17-18, the phrase of "which are mostly externally mixed" is not clear, and hence suggested to be changed as "and the particles were mostly mostly externally mixed". Line 20-21, the comma before which should be deleted, because the phrase is used for modifying the EP-4. Line 21-22, "industry-induced haze (EP-1) and biomass burning-induce haze (EP-5)" is suggested to be changed as "the industry-induced pollution episode (EP-1) and biomass burning-induce pollution episode (EP-5) ". Line 22-24, The two sentences seemed to be independent, lack of logic, and thus, the two sentences are suggested to be replaced by "Compared with the EP-2 and EP-4, the AOD values and the size distribution of particles during EP-1 and EP-5 were much greater because of relatively high particle concentrations ". Line25-26, the sentence was suggested to be replaced by "In contrast to the EP-1, a large fraction of soot which sticks to KCl, sulphate or nitrate particles was detected during the EP-5", implying the evident influence of severe crop residue combustion. Line 26-28, the sentence was suggested to be replaced by "Additionally, evident enhancement of light absorption was observed during the EP-5, which was mainly ascribed to both BC acceleration and other absorbing substances". Line 28-31, the sentences are better replaced by "However, soot was found mostly internally mixed with sulphate and nitrate during a soot fog episode (EP-3), resulting in evident enhancement of light absorption". 4.Introduction: Line 44-47, any kind of particles in the atmosphere have scattering effect, why did you only stress on inorganic salts and light-color organic carbon? The sentences is better replaced by "inorganic salts and light-color organic carbon have a "cooling effect"

on climate due to decreasing permeation of solar irradiation onto the Earth's surface through solely scattering sun light". There are still some sentences in the section being needed to be improved. 5.Results and discussion: Line 266-267, the sentence of "the north wind was relatively clean and the time was insufficient for a heavy accumulation" is not proper and clear. Wind can be only described by speed and direction, and hence the sentence is better replaced by "the air parcel from the North was relatively clean". What's "the time" in the latter half sentence"? Line 282, the title of "Optical parameter variation" is better replaced by "The variation of aerosol optical characters". Line 324, the title of "TEM analysis" is suggested to be "Morphology and chemical composition of aerosols". Line 319: "Especially when air masses moved from south direction to the sampling site aerosols were influenced by heavy soot-sulfate-OC-mixed pollution". How did you draw the conclusion? Line 388: Are you sure haze and fog episodes had a high possibility of collision just due to the heavy particle loading? You should add relevant reference to confirm your deduction. Line 390, the title of "Optical properties related to morphological of aerosols" is better replaced by "the relationship between the aerosol optical properties and morphologies "
* * *

---

## Referee Comment (RC3) · Anonymous Referee #3 · 18 Feb 2017

This paper has discussed the characteristics of aerosol optical properties, morphologies and their relationship in urban Beijing during the clear, haze and fog episodes, based on the analysis on the samples from 24th May to 22nd Jun, 2012. The state-of-the-art instrument has been used including Transmission Electron Microscope (TEM), in combination of a Cavity Ring Down Spectrometer (CRDS), a nephelometer, and an aethalometer. Five episodes were categorized according to the meteorological conditions, composition and optical variation. Results are interesting and should be published in the ACP journal.

I have checked the whole text of the manuscript and have made the following suggestions:

1. Line 259-260: The time range mentioned here is from 28th May to 29th May, but in Figure 1, it is from 24th May to 29th May. Pleas check and keep in consistency. 2. Line 284: How do you obtain the value R=0.603. 3. Line 385-389: There might be a misunderstanding of the definition of the internal and external mixing stages. The adjacent particles belong to the category of "inhomogeneous" internal mixing. Please refer to the relative papers for the definition. 4. Line 390: the title of this section may be changed into "Optical properties related to morphological types of aerosols". 5. Line 700: The meaning of "No." in Table 1 is not clear. 6. Line 710-713: It seems this is not Figure 3, instead, it may be Figure 6. Similarly, Figure 5 in Line 717-720 might be Figure 3, and Figure 6 in Line 721-722 might be Figure 5. Please check this section. 7. Line 726 Figure 1: The keys for this diagram are not very clear. The upper one: in addition to rain, fog, and haze days, the clear days should be expressed in white color key. The middle one: what are the meanings of the grey color and orange color? 8. Line 738 Figure 2: Keys for this diagram should be added. What are represented by those different colors of lines? 9. Line 748 Figure 3: Keys in figure 3c are erroneously used. The figure is not consistent with the description in text of Line 187-188. 10. Line 761 Figure 5: The values in the vertical axis should be 20, 40, 60, 80, and 100 percentages. Besides, are the percentages in this diagram based on statistics of the area or number? What about the values of the rainy days? 11. Line 766 Figure 6: The types in the classification shown in this figure are not consistent with those in Figure 4. The mineral particle type is missing in Figure 6, and still in this figure, the values of the rainy days are missing as well. 12. The keys in Figure 1, 3, and 7 should include those od the clear days (for those white areas). Also, the data for the clear days should be added.
* * *

---

## Author Comment (AC1) · 19 Feb 2017

Dear editor: Here we submit our revised manuscript for consideration to be published on Atmospheric chemistry and physics. The further information about our manuscript is as follows: Topic: Real-Time Aerosol Optical Properties, Morphology and Mixing States under Clear, Haze and Fog Episodes in the summer of Urban Beijing Type of Manuscript: article Authors: Rui Li1, Yunjie Hu1, Ling Li1, Hongbo Fu1,2,*, Jianmin Chen1,* Corresponding author: Hongbo Fu; Address: Department of Environmental Science and Engineering, Fudan University, Shanghai 200433, China; Tel.: (+86)21-5566-5189; Fax: (+86)21-6564-2080; Email: fuhb@fudan.edu.cn. Jianmin Chen; Address: Department of Environmental Science and Engineering, Fudan University, Shanghai 200433, China; Tel.: (+86)21-5566-5189; Fax: (+86)21-6564-2080; Email: jmchen@fudan.edu.cn

Firstly, we acknowledge the comments of anonymous reviewers, and are also grateful to the efficient serving of the editor. We have already revised MS based on the reviewers' comments. We also inspected MS roundly and corrected some errors in English presentation. We are sure that the revised MS adhere to Science of the Total Environment format. The marked MS was also uploaded to be easily reviewed.

Response 1. The English thorough the manuscript have been improved by an English native speaker 2. Line 12: Indeed, an overview of the study or its objectives should be provided in the abstract. Thus, "Aerosol particles play significant roles on the climate-forcing agent via its optical absorption properties. However, the relationship between characteristics of aerosol particles and optical absorption remains poorly understood" has been added in the abstract. 3. Line 30: "Soot fog period" means the fog episode filled with large amount of soot. 4. Line 289: "R" has been changed into "r". Thank you for reviewer's chariness. 5. Line 373: "organic matter" is a right expression, which has much difference from soot and tar ball. Li et al. (2009) and Fu et al. (2012) also used "organic matter" expression in their paper. Organic matter generally exhibits amorphous structure with abundant carbon and minor oxygen, whereas soot are chain-like aggregates of carbon-bearing spheres. Tar ball displays spherical appearance with substantial carbon, while they generally do not possess other elements. Organic matter could be released via biomass burning, vehicle exhausts, and other human activities by directly process. Besides, they can also be generated through photochemical reaction between VOCs and some photochemical oxidants such as $O_3$ and $NO_x$. 6. Line 407: The cubic shape of K-rich particles suggested they have not undergone long transportation or severe photochemical reaction because cubic K-rich particles were mainly produced from the molten nature of the material at high temperatures. The K-rich particles undergoing the long-range transport are generally encapsulated by visible coatings. 7. Indeed, coarser particles were near the centers of the grid and

finer particles on the periphery. However, three to four areas were chosen from the center and periphery of the sampling spot on each grid in order to ensure that the analyzed particles were representative. Therefore, we can assure the quality of the data. 8. Much literature published after 2016 about optical properties has been citied.
* * *

---

## Author Comment (AC3) · 19 Feb 2017

Firstly, we acknowledge the comments of anonymous reviewers, and are also grateful to the efficient serving of the editor. We have already revised MS based on the reviewers' comments. We also inspected MS roundly and corrected some errors in English presentation. We are sure that the revised MS adhere to Science of the Total Environment format. The marked MS was also uploaded to be easily reviewed.

Response 1. The English thorough the manuscript have been improved by an English native speaker 2. Many new published paper have been added into the manuscript. 3. Line 16: The sentences have been changed into "Aerosol optical properties and morphologies were measured by TEM, CRDS, a nephelometer and an aethalometer

in a urban site of Beijing from 24 May to 22 June". 4. Line 17: Indeed, the sentence is meaningless and has been deleted. 5. Line 23:"which" was replaced by "and the particles". The comma has been deleted 6. Line 27: the sentences has been changed into "industry-induced haze (EP-1) and biomass burning-induced haze (EP-5) were both affected by the south air mass". 7. Line 27-28: The two sentences have been replaced by "Compared with the EP-2 and EP-4, the AOD values and the size distribution of particles during EP-1 and EP-5 were much greater because of relatively high particle concentrations." 8. Line 32-35: The sentence was replaced by "In contrast to the EP-1, a large fraction of soot which sticks to KCl, sulphate or nitrate particles was detected during the EP-5". 9. Line 35: The sentence was replaced by "Additionally, evident enhancement of light absorption was observed during the EP-5, which was mainly ascribed to both BC acceleration and other absorbing substances". 10. Line 39: The sentence has been changed into "However, soot was found mostly internally mixed with sulphate and nitrate during a soot fog episode (EP-3), resulting in evident enhancement of light absorption". 11. Line 57-58: Indeed, the sentence has been changed into "inorganic salts and light-color organic carbon have a cooling effect". 12. Line 280: The sentence was changed into "the air parcel from the North was relatively clean". 13. Line 295: The sentence was changed into "The variation of aerosol optical characters" 14. Title 3.3: The title was replaced by "Morphology and chemical composition of aerosols" 15. Line 334: The conclusion was drawn because some previous studies have confirmed that soot, organic matter, and sulfates were generated from the industrial activities, domestic cooking, and biomass burning. Many industrial activities and biomass burning have been observed in South China. 16. Line 404: Haze and fog episodes generally had a high possibility of collision, which was caused by heavy particle loading. In addition, prolonged remaining of heavy particles was also a factor leading to the collision. Many relevant references have been added in the manuscript. 17. Line 407: The title was replaced by "the relation of optical properties and the morphologies of aerosol particles".

[Figure]

Please also note the supplement to this comment:
http://www.atmos-chem-phys-discuss.net/acp-2016-976/acp-2016-976-AC3-supplement.pdf
* * *

---

## Author Comment (AC4) · 19 Feb 2017

Firstly, we acknowledge the comments of anonymous reviewers, and are also grateful to the efficient serving of the editor. We have already revised MS based on the reviewers' comments. We also inspected MS roundly and corrected some errors in English presentation. We are sure that the revised MS adhere to Science of the Total Environment format. The marked MS was also uploaded to be easily reviewed.

Response 1. The English thorough the manuscript have been improved by an English native speaker 2. Line 12: Indeed, an overview of the study or its objectives should be provided in the abstract. Thus, "Aerosol particles play significant roles on the climate-forcing agent via its optical absorption properties. However, the relationship

between characteristics of aerosol particles and optical absorption remains poorly understood" has been added in the abstract. 3. Line 30: "Soot fog period" means the fog episode filled with large amount of soot. 4. Line 289: "R" has been changed into "r". Thank you for reviewer's chariness. 5. Line 373: "organic matter" is a right expression, which has much difference from soot and tar ball. Li et al. (2009) and Fu et al. (2012) also used "organic matter" expression in their paper. Organic matter generally exhibits amorphous structure with abundant carbon and minor oxygen, whereas soot are chain-like aggregates of carbon-bearing spheres. Tar ball displays spherical appearance with substantial carbon, while they generally do not possess other elements. Organic matter could be released via biomass burning, vehicle exhausts, and other human activities by directly process. Besides, they can also be generated through photochemical reaction between VOCs and some photochemical oxidants such as O3 and NOx. 6. Line 407: The cubic shape of K-rich particles suggested they have not undergone long transportation or severe photochemical reaction because cubic K-rich particles were mainly produced from the molten nature of the material at high temperatures. The K-rich particles undergoing the long-range transport are generally encapsulated by visible coatings. 7. Indeed, coarser particles were near the centers of the grid and finer particles on the periphery. However, three to four areas were chosen from the center and periphery of the sampling spot on each grid in order to ensure that the analyzed particles were representative. Therefore, we can assure the quality of the data. 8. Much literature published after 2016 about optical properties has been citied.

Please also note the supplement to this comment:
http://www.atmos-chem-phys-discuss.net/acp-2016-976/acp-2016-976-AC4-supplement.pdf

**Supplement:**

| 1  | Real-Time Aerosol Optical Properties, Morphology and Mixing                                                                    |  |  |  |  |  |  |  |
|----|--------------------------------------------------------------------------------------------------------------------------------|--|--|--|--|--|--|--|
| 2  | States under Clear, Haze and Fog Episodes in the summer of Urban                                                               |  |  |  |  |  |  |  |
| 3  | Beijing                                                                                                                        |  |  |  |  |  |  |  |
| 4  | Rui Li 1 , Yunjie Hu 1 , Ling Li 1 , Hongbo Fu 1,2,* , Jianmin Chen 1,* |  |  |  |  |  |  |  |
| 5  | 1 Shanghai Key Laboratory of Atmospheric Particle Pollution and Prevention, Department of                           |  |  |  |  |  |  |  |
| 6  | Environmental Science & Engineering, Fudan University, Shanghai 200433, China Chinese                                          |  |  |  |  |  |  |  |
| 7  | Academy of Sciences, Institute of Atmosphere Physics, Beijing 100029.                                                          |  |  |  |  |  |  |  |
| 8  | 2 Collaborative Innovation Center of Atmospheric Environment and Equipment Technology                               |  |  |  |  |  |  |  |
| 9  | (CICAEET), Nanjing University of Information Science and Technology, Nanjing 210044,                                           |  |  |  |  |  |  |  |
| 10 | China                                                                                                                          |  |  |  |  |  |  |  |
| 11 | Abstract                                                                                                                       |  |  |  |  |  |  |  |
| 12 | Aerosol particles play significant roles on the climate-forcing agent via its optical absorption                               |  |  |  |  |  |  |  |
| 13 | properties. However, the relationship between characteristics of aerosol particles and optical                                 |  |  |  |  |  |  |  |
| 14 | absorption remains poorly understood. Characteristics of aerosol optical properties,                                           |  |  |  |  |  |  |  |
| 15 | morphologies and their relationship were studied in urban Beijing during the clear, haze and                                   |  |  |  |  |  |  |  |
| 16 | fog episodes, sampled from 24th May to $22_{nd}$ Jun, 2012. Transmission Electron Microscope                                   |  |  |  |  |  |  |  |
| 17 | (TEM), a Cavity Ring Down Spectrometer (CRDS), a nephelometer and an aethalometer were                                         |  |  |  |  |  |  |  |
| 18 | employed to investigate the corresponding changes of the aerosol properties. Five episodes                                     |  |  |  |  |  |  |  |
| 19 | were categorised according to the meteorological conditions and composition. The results                                       |  |  |  |  |  |  |  |
| 20 | showed that the clear episode (EP-2 and EP-4) featured as the low Aerosol Optical Depth (AOD                                   |  |  |  |  |  |  |  |
| 21 | = 0.72) and less pollutants compared with haze (1.14) and fog (2.92) episodes, which are mostly $1$                            |  |  |  |  |  |  |  |

| 22 | externally mixed. The high Ångström exponent (> 2.0) suggests that coarse particles were             |
|----|------------------------------------------------------------------------------------------------------|
| 23 | scarcely observed in EP-2 due to the washout of a previous heavy rain, whereas they were             |
| 24 | widespread in EP-4 (Ångström exponent = $0.04$ ), which had some mineral particles introduced        |
| 25 | from the north. In contrast, industry-induced haze (EP-1) and biomass burning-induced haze           |
| 26 | (EP-5) were both affected by the south air mass. Higher AOD values illustrated heavy loading         |
| 27 | particle concentrations. All of the particles were classified into nine categories including S-rich, |
| 28 | N-rich, mineral, K-rich, soot, tar ball, organic, metal and fly ash on the basis of TEM analysis.    |

[revised manuscript text omitted]
  $\omega$  is 323 lower in this study, suggesting that more soot is uploaded into the atmosphere during this period. It is well 324 known that soot emission is much higher in the past years, mainly contributed by the residential coal 325 combustion, biomass burning, coke production, and diesel vehicles (Wang et al., 2012b). Especially, when 326 air masses moved from south direction the sampling site were influenced by heavy polluted air mass mixed 327 by soot, sulfate, and OC-components, from the dense population centres and industrial areas (Sun et al., 328 2006; Wang et al., 2006), which was also confirmed by the TEM observation.

329 3.3 TEM analysis

Based on morphology and chemical composition, 1173 particles were classified into nine categories: S-rich
(Fig. 4a), N-rich (Fig. 4b), mineral (Fig. 4c), K-rich (Fig. 4d), soot (Fig. 4e), tar ball (Fig.4f), organic
(Fig.4g), metal (Fig. 4h) and fly ash (Fig. 4i). The classification is similar to the work reported by Li and
Shao (2009).

334 The most common particles are sulphates and nitrates (Figs. 4a and b), which are of the size around 1.0 335 μm, and have a light scattering ability (Jacobson, 2001). Sulphates appeared as subrounded masses under 336 the TEM, which decomposed or evaporated under the electron beam exposure. Conventionally, they were 337 formed by the reaction of precursor  $SO_2$  or  $H_2SO_4$  with other gases or particles (Khoder, 2002). Nitrates 338 were mostly of scalloped morphology in the TEM images. They were relatively stable when exposed to the 339 electron beam. Nitrates formed through the homogeneous reaction with the precursor either NO2 or 340 heterogenic reaction with HNO3 (Khoder, 2002). (Pathak et al., 2004; Seinfeld and Pandis, 2012). 341 In the clear days, as the result of effects of northern air mass, dust particles were relatively abundant. The

size of dust particles (Fig. 4c) were large, usually bigger than 1.0 µm, so far as to 8.0 µm. Their compositions
differed from each other, mostly are silicates and calcium sulphate or carbonate, all of which were stable
under the exposure of the electron beam. Dust particles were reported to have a light scattering effect,

resulting in a negative aerosol radiative forcing (Wang et al., 2009b). They took up a large portion in EP-4,
impacted by the north wind taking along particles from the dusty regions.

347 As for the haze episode, K-rich particles (Li et al., 2010; Duan et al., 2004; Engling et al., 2009), soot (Li et al., 2010), tar ball (Chakrabarty et al., 2010;Bond, 2001) and organic (Lack et al., 2012) were more 348 349 observed under the TEM. K-rich particles (Fig. 4d) often existed as sulphate or nitrate. A larger fraction of 350 K-rich particles was observed in EP-5 than those in the other periods. Together with the back trajectories 351 and fire spot maps, it was supposed that the regional haze occurred in EP-5 was contributed significantly 352 by the biomass burning. K-rich particles were characterized by the irregular shape, which was unstable 353 when exposed to electron beam. KCl was barely detected in the samples, even though it has been 354 recommended that KCl was internally mixed with K2SO4 and KNO3 in fresh biomass burning plumes (Li 355 et al., 2010;Li et al., 2003;Adachi and Buseck, 2008). Based on the EDS data, K-rich particles in the present 356 work mostly consisted of N, Na, O, S, and K, whereas it was free of Cl, implying KCl could have suffered 357 from chemical reactions and transformed into sulphates or nitrates (Li and Shao, 2010). Such particles 358 displayed a negative climate forcing (Hauglustaine et al., 2014).

359 It was well documented that soot (Fig. 4e) was vital to light absorption, which could alter regional 360 atmospheric stability and vertical motions, the large scale circulation and precipitation with significant 361 regional climate effects (Ramanathan et al., 2001; Jacobson, 2002). It was well characterized by a structure 362 like onion ring, resembling a fractal long chain as agglomerates of small spherical monomers (Li and Shao, 363 2009). The fresh soot was loose and externally mixed. However, after undergoing a long-range 364 transportation and aging in the atmosphere, soot became more compacted, with a slight increase of O 365 concentration because of the photochemistry (Stanmore et al., 2001; Krasowsky et al., 2016). Meanwhile, 366 soot generally attached to other particles on the surface or serves as the core for other particle formation. 367 Tar ball (Fig. 4f) was present as a spherical carbon ball with a small fraction of O. It was thought to 368 origin from the smouldering combustion and have relatively strong absorption effects (Chakrabarty et al., 369 2010; Bond, 2001). Tar balls constituted a large fraction of the fresh emitted wildfire carbonaceous particles 370 (China et al., 2013;Lack et al., 2012). But it was seldom observed in the present work, even in EP-5 when

there was severe biomass burning emission, which may be due to the difference in burning species andconditions.

Organic matter (Fig. 4g) identified by HRTEM was amorphous species, and was stable under the strong electron beam exposure. It could be traced to the direct emission such as biomass burning (Lack et al., 2012), or the second reaction between VOCs with ozone (Wang et al., 2012a). It can absorb radiation in the low-visible and UV wavelengths (Chakrabarty et al., 2010;Clarke et al., 2007;Lewis et al., 2008;Hoffer et al., 2006). In addition, when compassing soot as the core, organic matter can enhance absorption by internal mixing (Adachi and Buseck, 2008).

For the common haze and fog episodes, the stagnated weather favours the accumulation of pollutants, especially metal particles and fly ash (Hu et al., 2015). Metal particles (Fig. 4h) were generally round and stable under the TEM. Fly ash (Fig. 4i) was a dark sphere with large size of  $> 1 \ \mu$  m. It was a common product of industrial activities in the northern China (Shi et al., 2003). As the complex refractive index (CRI) indicated, metal oxide particles and fly ash can scatter light, but the former has a weak absorption ability while the later has almost no light absorption ability (Ebert et al., 2004).

385 Figure 5 shows percentage of nine components in clear, haze and fog episodes under external mixing, 386 internal mixing and adjacent states (partially internal mixing). About 28% of particles were internally mixed 387 in the foggy days, while about 52% of particles exhibited external mixing state in clear days based on the 388 TEM analysis. Mineral particles were inclined to be externally mixed with K-rich particles and organic 389 matter in clear days, while the external ratio of other particles were relatively lower, particularly in the haze 390 and fog days. Li et al. (2010) showed that mineral particles generally displayed external association with 391 organic matter or other particles. However, many fine particles including metal-bearing particles, fly ash 392 and soot were often internally mixed with S-rich and K-rich particles, particularly during the fog-haze 393 episodes. Shi et al. (2008) reported that rapid aging of fresh soot tended to appear during the fog-haze days, 394 which were generally associated with ammonium sulfate. Heavy polluted air generally promoted the 395 coagulation between S/K-rich particles and those fine particles such as metal particles, soot, and fly ash (Li and Shao, 2009), which could explain the results. Additionally, haze and fog episodes held a higher possibility of collision and attachment due to the heavy particle loading and prolonged remaining in the atmosphere, leading to a higher internal mixed state percentage around 65%.

**399 3.4 The relation of optical properties and the morphologies of aerosol particles**

400 The different morphologies of the particles collected from the different weather can be easily identified 401 under the TEM, as shown in Fig. 6. Due to the washout effect of the heavy rain, the particles collected in 402 the typical clear period of EP-2 were much smaller in size (Figs. 6a, b), which was in good agreement with 403 the larger Ångström exponent. The coarse particles, such as dusts, were hardly observed, whereas a few K-404 rich particles were detected, of which presented in small cubic shape. Such particles could be explained by 405 the coal combustion around the sampling site due to the slight fire spots presence. Besides, the cubic shape 406 of K-rich particles suggested they have not undergone long transportation or severe photochemical reaction 407 because cubic K-rich particles were generally generated from the molten nature of the material at high 408 temperatures (Ault et al., 2012). Likewise, soot was generally less oxidized in the EP2 periods, maintaining 409 fractional morphologies and externally mixed. Small metal particles and amorphous Zn-particles dominated 410 the fine particles, which was ascribed to the industrial activity and/or waste incineration (Choël et al., 411 2006;Moffet et al., 2008).

412 In the EP-5 episode, the increased aerosol loading played a remarkable role in the enhancement of 413 scattering coefficient and decrease of visibility (Kang et al., 2013;Charlson et al., 1987;Deng et al., 2008). 414 Because of the high rate of aerosol collision, particles were larger than those in the clear days (Figs. 6c, d), 415 leading to a smaller Ångström exponent. Almost of the soot particles observed under the TEM were 416 compact and adhesive. It was internally mixed with the K-rich particles, which were larger, rounder or with 417 a coating of high S components. As discussed above, they were probably transported from the south crop 418 residual burning and undergo the ageing in the atmosphere, confirmed by the trajectories passing through 419 intense fire spots. Due to the high concentration of soot, EP-5 were characterized by a high absorption 420 coefficient, shown in Fig. 3.

421 The BC variations in the different weather types during the sampling period were illustrated in Fig. 7. 422 The preliminary component of BC could be viewed as the soot. High BC concentration was easily 423 recognized in EP-5 with a mean value of 12.8 µg m-3, while it is low up to 1.04 µg/m3 during the clear 424 periods. The former is about 11.3 times higher than that of the latter, which is due to the lower boundary 425 layer- In comparison, absorption coefficient of EP-5 (468.7 Mm-1) was about 94.7 times higher than that of 426 EP-4 (1.3 Mm-1), more than 8 times of the BC ratio. It was supposed that BC was internally mixed with 427 other aerosols in the EP-5, which lead to the considerable elevation of absorption coefficient (Tan et al., 428 2016). However, Models models estimated an enhancement of BC forcing up to a factor of 2.9 when BC is 429 internally mixed with other aerosols, compared with externally mixed scenarios (Jacobson, 2001), which 430 was much lower than this case. Accordingly, other light absorbing substances may contribute to the 431 discrepancy. For example, Brown brown carbon is an indispensable component of biomass burning, which 432 has a strong absorption ability as well (Hoffer et al., 2006;Andreae and Gelencsér, 2006). Other particles 433 like-such as dust may also contribute to the over-enhanced absorption coefficient (Yang et al., 2009). Our 434 observations were agreement with the previous studies reported by (Wang et al., 2009b; Xia et al., 2006), 435 which shows that aerosol particles under hazy weather conditions generate a positive heating effect on the 436 atmospheric eolumn (Wang et al., 2009b; Xia et al., 2006).

437 In the foggy days of EP-3 episode, the high  $PM_{10}$  concentration and AOD caused significant increase of 438 scattering coefficient (Tan et al., 2016). Furthermore, metal-bearing particles and soot were internally 439 associated with some coatings including S-rich, N-rich and K-rich particles. Zhang et al. (2008) reported 440 that coating with sulphuric acid enhance the optical properties of soot aerosols. Furthermore, the collected 441 particles displayed larger size than those collected from the clear days under the TEM (Figs. 6e, f). The 442 larger size particles in the foggy days could be caused by hydroscopic growth under the high relative 443 humidity, and the collision among the overloading particles, which was likewise illustrated by the Ångström 444 exponent shown in Fig.3. Consequently, the larger particles enhance the scattering of sunlight, and lead to 445 more apparent impairment of visibility (Quan et al., 2011). Chow et al. (2002a) reported that RH also has 446 a profound impact on visibility. Some fan-like nitrate particles have inclusions which may act as the growth

447 cores or be encompassed during the hydroscopic growth. Bian et al. (2009) reported that whenever the RH 448 is elevated, its importance to AOD is substantially amplified if the particles are hygroscopic (Bian et al., 449 2009). Li et al. (2010) found that soot particles became hydrophilic when they were coated with the water-450 soluble compounds such as sulphates or nitrates, implying that soot can provide important nuclei for the 451 development of aerosol particles. Furthermore, Fig. 6e and f illustrate a large fraction of internally mixed 452 soot. It was not visible until being exposed to electron beam for a short time. As for an internally mixed 453 particle, sulphate and nitrate coatings act as a "focusing mirror", and enhanced light absorption greatly. 454 Therefore, the BC concentration in foggy conditions was 6.12 µg m3, and the absorption coefficient is 143.7 455 Mm-1, which were 2.09 and 0.83 times of the hazy days, respectively. Model calculation also have 456 recommended that light absorption ability of the internally mixed soot particles were enhanced by 30% 457 than that of soot alone (Fuller et al., 1999). A variety of metal particles were also observed in the foggy 458 days, as foggy days had a stable low upper layer boundary and slight wind, leading to the accumulation of 459 pollutions. These pollution sources range from steel plants and waste incineration to vehicle emission and 460 so on (Hu et al., 2015).

**461 4 Conclusions**

462 The relationship between characteristics of aerosol particles and optical properties is of importance to 463 the atmospheric chemistry research. However, the relationship between characteristics of aerosol particles 464 and optical absorption remains poorly understood. Characteristics of aerosol optical properties, 465 morphologies and their relationship were studied in urban Beijing during the clear, haze and fog episodes, 466 sampled from 24th May to 22nd Jun, 2012. Transmission Electron Microscope (TEM), a Cavity Ring Down 467 Spectrometer (CRDS), a nephelometer and an aethalometer were employed to investigate the corresponding 468 changes of the aerosol properties. Five episodes were categorised according to the meteorological 469 conditions and composition. The results indicated that the clear episode (EP-2 and EP-4) was characterized 470 as the low aerosol Optical Depth (AOD = 0.72) and less pollutants compared with haze (1.14) and fog (2.92) 471 episodes, which are mostly externally mixed. The high Ångström exponent (> 2.0) suggests that coarse 472 particles were scarcely observed in EP-2 due to the washout of a previous heavy rain, whereas they were 473 widespread in EP-4 (Ångström exponent = 0.04), which had some mineral particles introduced from the 474 north. In contrast, industry-induced haze (EP-1) and biomass burning-induced haze (EP-5) were both 475 affected by the south air mass. Higher AOD values illustrated heavy loading particle concentrations. All of 476 the particles were classified into nine categories including S-rich, N-rich, mineral, K-rich, soot, tar ball, 477 organic, metal and fly ash based on the TEM analysis. In the haze episode, as the influence of severe crop 478 residue combustion, a large fraction of soot was detected, which sticks to sulphate or nitrate particles 479 transformed from KCl. Both black carbon (BC) acceleration, internally mixed effects, and other light 480 absorbing substances, contributed the light absorption enhancement. For foggy days, soot was mostly 481 internally mixed with sulphates and nitrates, which revealed themselves after electron exposure under the 482 TEM. The larger size distribution was likely to be caused by both hygroscopic growth and collision between 483 particles during the aging. About 28% of particles were internally mixed in the foggy days, which favored 484 the light absorption. The comparison of all the episodes provides a deeper insight of how mixing states 485 influence the aerosol extinction properties and also a clue to the air pollution control in the crop burning 486 seasons. The result presented herein is beneficial to air pollution control and prevention in China.

**487 ACKNOWLEDGMENTS**

[revised manuscript text omitted]

| Sampling Time (BST a ) |          |             |                   | RH  | Temp. | Wind           |           | Visibility |     |  |
|-----------------------------------|----------|-------------|-------------------|-----|-------|----------------|-----------|------------|-----|--|
| Date                              | Starting | Duration    | Conditions        | (%) | (°C)  | Speed
(m/s) | Direction | (km)       | No. |  |
| 25-05-                            | 13:40    | 4 min       | Clear             | 20  | 29    | 2              | 160       | -          | 136 |  |
| 2012                              | 15.10    | 1           | citur             | 20  | 2)    | 2              | 100       |            | 150 |  |
| 30-05-                            | 0.21     | 16          | Chara             | 20  | 24    | 7              | 250       |            | 02  |  |
| 2012                              | 9:31     | 16 min      | Clear             | 29  | 24    | /              | 350       | -          | 92  |  |
| 02-06-                            | 0.00     |             |                   |     | 20    |                | 100       |            |     |  |
| 2012                              | 9:00     | 1 min       | Mist              | 83  | 20    | 4              | 180       | 2          | 146 |  |
| 02-06-                            | 13:27    |             |                   |     |       |                |           |            |     |  |
| 2012                              |          | 2 min       | Clear             | 48  | 27    | 4              | 190       | -          | 138 |  |
| 03-06-                            | 10:13    |             |                   |     |       |                |           |            |     |  |
| 2012                              |          | 15 s        | Fog               | 88  | 22    | 1              | variable  | 1.2        | 110 |  |
| 18-06-                            |          |             |                   |     |       |                |           |            |     |  |
| 2012                              | 18:52    | 52 2 min Ha | Haze              | 55  | 29    | 3              | 140       | 3          | 172 |  |
| 19-06-                            | 9:10     |             |                   |     |       |                |           |            |     |  |
| 2012                              |          | 2 min Haz   | Haze              | 61  | 25    | 1              | variable  | 2.8        | 120 |  |
| 21-06-                            | 9:10     | 5-          |                   |     |       |                |           |            |     |  |
| 2012                              |          | 1 min Haze  | Haze              | 69  | 26    | 2              | 110       | 2.2        | 117 |  |
| 23-06-                            |          |             |                   |     |       |                |           |            |     |  |
| 2012                              | 12:45    | 2 min       | Mist b | 84  | 25    | 4              | 120       | 3          | 142 |  |

the number of particle analysed in each sample.

aBeijing standard time (8 h prior to GMT).

bMist is studied here as fog.

724

**726 Figure captions**

- 727 Figure 1. 5 episodes categorization. EP-1 features haze induced mainly from transportation of south
- 728 industrial pollution, EP-2 clear, EP-3 frequent transition among haze, fog and clear conditions, EP-4 clear
- 729 with rain interrupted, and EP-5 haze resulted mainly from the biomass burning.
- Figure 2. The 3-day back-trajectory clusters of each episode, arriving at Beijing at the height of 100 m,
- 731 together with the fire spot distribution of these periods.
- 732 Figure 3. TEM typical views of the particles in clear (upper panel), haze (middle panel) and fog episodes
- 733 (bottom panel). 9 components are marked with the colourful arrows. (a1) (b1) (c1) (d1) (e1) (f1) is obtained
- before the electron exposure and (a2) (b2) (c2) (d2) (e2) (f2) is after exposure. A fraction of S-rich particles
- and other unstable particles decompose after electron exposure.
- 736 Figure 4.9 categories of particles under the TEM view. The inserted spectra are obtained by the EDS, and
- the grid like images are acquired from the SAED. (a) S-rich, (b)N-rich, (c)mineral. (d)K-rich, (e)soot, (f)tar
- 738 ball, (g)organic, (h)metal, (i)fly ash.
- 739 Figure 5. Variation of optical parameters during the study period. (a) Total Aerosol optical depth (AOD),
- 740 and AOD resulted from Mie scatter and Rayleigh scatter; (b) Ångström exponent (å) computed from the
- pairs of 700 nm and 450 nm, 700 nm and 550 nm, and 550 nm and 450 nm; (c) light extinction, absorption
- and scattering coefficients; (d) calculated single scattering albedo (SSA).
- Figure 6. Percentages of 9 particle components under clear, haze and fog conditions with different mixingstates.
- 745 Figure 7. BC concentrations converted from the data measured by AE-31 and MAAP. Good correlation is
- 746 observed.
- 747

---

## Author Comment (AC5) · 6 Mar 2017

Firstly, we acknowledge the comments of anonymous reviewers, and are also grateful to the efficient serving of the editor. We have already revised MS based on the reviewers' comments. We also inspected MS roundly and corrected some errors in English presentation. We are sure that the revised MS adhere to Atmospheric Chemistry and Physics. The marked MS was also uploaded to be easily reviewed.

Comment 1: Line 259-260: The time range mentioned here is from 28th May to 29th May, but in Figure 1, it is from 24th May to 29th May. Please check and keep in consistency. Response: Line 262: The time range mentioned here is from 24 th May to 29 th May after the careful check. Thus, the "28 th" has been changed into

"24 th". Thank you for reviewer's chariness. Comment 2: Line 284: How do you obtain the value R=0.603. Response: Line 286: The R value was obtained using Pearson correlation analysis. The R value is the correlation coefficient between AOD and PM10 with a 95% confidence interval. Comment 3: Line 385-389: There might be a misunderstanding of the definition of the internal and external mixing stages. The adjacent particles belong to the category of "inhomogeneous" internal mixing. Please refer to the relative papers for the definition. Response: Line 382-396: Indeed, adjacent particles belong to the category of inhomogeneous internal mixing after my critical review of references. Therefore, the "internal" and "adjacent" were replaced by "internal" and "internal (adjacent)" in Fig. 5, respectively. Comment 4: Line 390: the title of this section may be changed into "Optical properties related to morphological types of aerosols". Response: Line 397: The title has been changed into "Optical properties related to morphological types of aerosols". Comment 5: Line 700: The meaning of "No." in Table 1 is not clear. Response: Table 1: The title has been replaced by "Sampling time and instantaneous meteorological state". Comment 6: Line 710-713: It seems this is not Figure 3, instead, it may be Figure 6. Similarly, Figure 5 in Line 717-720 might be Figure 3, and Figure 6 in Line 721-722 might be Figure 5. Please check this section. Response: Fig 3-6: The figure caption is confused, and right figure caption was added in the manuscript. Comment 7: Line 726 Figure 1: The keys for this diagram are not very clear. The upper one: in addition to rain, fog, and haze days, the clear days should be expressed in white color key. The middle one: what are the meanings of the grey color and orange color? Response: Fig. 1: In the upper diagram, the color column has been added. In the middle one, brown, green, and orange color meant the haze, clear, and fog conditions, respectively, which were added in the figure caption. Comment 8: Line 738 Figure 2: Keys for this diagram should be added. What are represented by those different colors of lines? Response: Fig. 2: Green, purple, red, and blue line denotes the air parcel with the height of 500, 1000, 2000, and 3000 m, which was added in the figure caption. Comment 9: Line 748 Figure 3: Keys in figure 3c are erroneously used. The figure is not consistent with the description in text of Line 187-188. Response: Fig. 3: I think the Fig. 3 is consistent with the description in text of Line 187-188, no contradictory description was observed. Comment 10: Line 761 Figure 5: The values in the vertical axis should be 20, 40, 60, 80, and 100 percentages. Besides, are the percentages in this diagram based on statistics of the area or number? What about the values of the rainy days? Response: Fig. 5: The percentage in this diagram was obtained based on statistics of number. The main aim of our study is to compare the optical properties and morphologies of particles among haze, fog, and clear, and then decipher the relationship between optical properties and morphologies. However, the optical property and morphology in the rainy days were not set as our main objectives. Thus, the percentage of particles in the rainy days were not included in Fig. 5. Comment 11: Line 766 Figure 6: The types in the classification shown in this figure are not consistent with those in Figure 4. The mineral particle type is missing in Figure 6, and still in this figure, the values of the rainy days are missing as well. Response: Fig. 6: the Ca-S particles and rod belong the same class. The Ca-S particles were changed into mineral and rod was replaced by Ca-S particles. The particles in the rainy days were not included in our study. Comment 12: The keys in Figure 1, 3, and 7 should include those of the clear days (for those white areas). Also, the data for the clear days should be added. Response: Fig. 1, 3, and 7: the white color has been added in the revised version.

Please also note the supplement to this comment:
http://www.atmos-chem-phys-discuss.net/acp-2016-976/acp-2016-976-AC5-supplement.pdf

**Supplement:**

**Real-Time Aerosol Optical Properties, Morphology and Mixing**

**States under Clear, Haze and Fog Episodes in the summer of Urban**

**Beijing**

Rui Li[1], Yunjie Hu[1], Ling Li[1], Hongbo Fu[1,2,*], Jianmin Chen[1,*]

*[1] Shanghai Key Laboratory of Atmospheric Particle Pollution and Prevention, Department of*

*Environmental Science & Engineering, Fudan University, Shanghai 200433, China Chinese*

*Academy of Sciences, Institute of Atmosphere Physics, Beijing 100029.*

*[2] Collaborative Innovation Center of Atmospheric Environment and Equipment Technology*

*(CICAEET), Nanjing University of Information Science and Technology, Nanjing 210044,*

*China*

**Abstract**

Aerosol particles play significant roles on the climate-forcing agent via its optical absorption properties. However, the relationship between characteristics of aerosol particles and optical absorption remains poorly understood. Aerosol optical properties and morphologies were measured by TEM, CRDS, a nephelometer and an aethalometer in a urban site of Beijing from

24 May to 22 June. Five episodes were categorised according to the meteorological conditions and composition. The results showed that the clear episode (EP-2 and EP-4) featured as the low Aerosol Optical Depth (AOD = 0.72) and less pollutants compared with haze (1.14)

and fog (2.92) episodes and the particles are mostly externally mixed. The high Ångström exponent (> 2.0) suggests that coarse particles were scarcely observed in EP-2 due to the washout of a previous heavy rain, whereas they were widespread in EP-4 (Ångström exponent

= 0.04), which had some mineral particles introduced from the north. In contrast, industry- induced haze (EP-1) and biomass burning-induced haze (EP-5) were both affected by the south air mass. Compared with the EP-2 and EP-4, the AOD values and the size distribution of particles during EP-1 and EP-5 were much greater because of relatively high particle concentrations. All of the particles were classified into nine categories including S-rich, N-rich, mineral, K-rich, soot, tar ball, organic, metal and fly ash on the basis of TEM analysis. In the haze episode, In contrast to the EP-1, a large fraction of soot, which sticks to KCl, sulphate or nitrate particles was detected during the EP-5 Additionally, evident enhancement of light absorption was observed during the EP-5, which was mainly ascribed to both BC acceleration and other absorbing substances. However, soot was found mostly internally mixed with sulphate and nitrate during a soot fog episode (EP-3), resulting in evident enhancement of light absorption 
[revised manuscript text omitted]
, suggesting that more soot is uploaded into the atmosphere during this period. It is well known that soot emission is much higher in the past years, mainly contributed by the residential coal combustion, biomass burning, coke production, and diesel vehicles (Wang et al., 2012b). Especially, when air masses moved from south direction the sampling site were influenced by heavy polluted air mass mixed by soot, sulfate, and OC-components, from the dense population centres and industrial areas (Sun et al.,

2006;Wang et al., 2006), which was also confirmed by the TEM observation.

**3.3 "Morphology and chemical composition of aerosols**

Based on morphology and chemical composition, 1173 particles were classified into nine categories: S-rich (Fig. 4a), N-rich (Fig. 4b), mineral (Fig. 4c), K-rich (Fig. 4d), soot (Fig. 4e), tar ball (Fig.4f), organic (Fig.4g), metal (Fig. 4h) and fly ash (Fig. 4i). The classification is similar to the work reported by Li and

Shao (2009).

The most common particles are sulphates and nitrates (Figs. 4a and b), which are of the size around 1.0

μm, and have a light scattering ability (Jacobson, 2001). Sulphates appeared as subrounded masses under the TEM, which decomposed or evaporated under the electron beam exposure. Conventionally, they were formed by the reaction of precursor $SO_2$ or $H_2SO_4$ with other gases or particles (Khoder, 2002). Nitrates were mostly of scalloped morphology in the TEM images. They were relatively stable when exposed to the electron beam. Nitrates formed through the homogeneous reaction with the precursor either $NO_2$ or heterogenic reaction with $HNO_3$ (Khoder, 2002). (Pathak et al., 2004; Seinfeld and Pandis, 2012).

In the clear days, as the result of effects of northern air mass, dust particles were relatively abundant. The size of dust particles (Fig. 4c) were large, usually bigger than 1.0 μm, so far as to 8.0 μm. Their compositions differed from each other, mostly are silicates and calcium sulphate or carbonate, all of which were stable under the exposure of the electron beam. Dust particles were reported to have a light scattering effect, resulting in a negative aerosol radiative forcing (Wang et al., 2009b). They took up a large portion in EP-4, impacted by the north wind taking along particles from the dusty regions.

As for the haze episode, K-rich particles (Li et al., 2010; Duan et al., 2004; Engling et al., 2009), soot (Li et al., 2010), tar ball (Chakrabarty et al., 2010;Bond, 2001) and organic (Lack et al., 2012) were more observed under the TEM. K-rich particles (Fig. 4d) often existed as sulphate or nitrate. A larger fraction of

K-rich particles was observed in EP-5 than those in the other periods. Together with the back trajectories and fire spot maps, it was supposed that the regional haze occurred in EP-5 was contributed significantly by the biomass burning. K-rich particles were characterized by the irregular shape, which was unstable when exposed to electron beam. KCl was barely detected in the samples, even though it has been recommended that KCl was internally mixed with $K_2SO_4$ and $KNO_3$ in fresh biomass burning plumes (Li et al., 2010;Li et al., 2003;Adachi and Buseck, 2008). Based on the EDS data, K-rich particles in the present work mostly consisted of N, Na, O, S, and K, whereas it was free of Cl, implying KCl could have suffered from chemical reactions and transformed into sulphates or nitrates (Li and Shao, 2010). Such particles displayed a negative climate forcing (Hauglustaine et al., 2014).

It was well documented that soot (Fig. 4e) was vital to light absorption, which could alter regional atmospheric stability and vertical motions, the large scale circulation and precipitation with significant regional climate effects (Ramanathan et al., 2001;Jacobson, 2002). It was well characterized by a structure like onion ring, resembling a fractal long chain as agglomerates of small spherical monomers (Li and Shao,

2009). The fresh soot was loose and externally mixed. However, after undergoing a long-range transportation and aging in the atmosphere, soot became more compacted, with a slight increase of O

concentration because of the photochemistry (Stanmore et al., 2001; Krasowsky et al., 2016). Meanwhile, soot generally attached to other particles on the surface or serves as the core for other particle formation.

Tar ball (Fig. 4f) was present as a spherical carbon ball with a small fraction of O. It was thought to origin from the smouldering combustion and have relatively strong absorption effects (Chakrabarty et al.,

2010; Bond, 2001). Tar balls constituted a large fraction of the fresh emitted wildfire carbonaceous particles (China et al., 2013;Lack et al., 2012). But it was seldom observed in the present work, even in EP-5 when there was severe biomass burning emission, which may be due to the difference in burning species and conditions.

Organic matter (Fig. 4g) identified by HRTEM was amorphous species, and was stable under the strong electron beam exposure. It could be traced to the direct emission such as biomass burning (Lack et al.,

2012), or the second reaction between VOCs with ozone (Wang et al., 2012a). It can absorb radiation in the low-visible and UV wavelengths (Chakrabarty et al., 2010;Clarke et al., 2007;Lewis et al., 2008;Hoffer et al., 2006). In addition, when compassing soot as the core, organic matter can enhance absorption by internal mixing (Adachi and Buseck, 2008).

For the common haze and fog episodes, the stagnated weather favours the accumulation of pollutants, especially metal particles and fly ash (Hu et al., 2015). Metal particles (Fig. 4h) were generally round and stable under the TEM. Fly ash (Fig. 4i) was a dark sphere with large size of $> 1$ μ m. It was a common product of industrial activities in the northern China (Shi et al., 2003). As the complex refractive index (CRI) indicated, metal oxide particles and fly ash can scatter light, but the former has a weak absorption ability while the later has almost no light absorption ability (Ebert et al., 2004).

Figure 5 shows percentage of nine components in clear, haze and fog episodes under external mixing, internal mixing and adjacent states (partially internal mixing). About 28% of particles were internally mixed in the foggy days, while about 52% of particles exhibited external mixing state in clear days based on the

TEM analysis. Mineral particles were inclined to be externally mixed with K-rich particles and organic matter in clear days, while the external ratio of other particles were relatively lower, particularly in the haze and fog days. Li et al. (2010) showed that mineral particles generally displayed external association with organic matter or other particles. However, many fine particles including metal-bearing particles, fly ash and soot were often internally mixed with S-rich and K-rich particles, particularly during the fog-haze episodes. Shi et al. (2008) reported that rapid aging of fresh soot tended to appear during the fog-haze days, which were generally associated with ammonium sulfate. Heavy polluted air generally promoted the coagulation between S/K-rich particles and those fine particles such as metal particles, soot, and fly ash (Li and Shao, 2009), which could explain the results. Additionally, haze and fog episodes held a higher possibility of collision and attachment due to the heavy particle loading and prolonged remaining in the atmosphere (Li and Shao, 2009; Li et al., 2010), leading to a higher internal mixed state percentage around

65%.

**3.4 Optical properties related to morphological types of aerosols**

The different morphologies of the particles collected from the different weather can be easily identified under the TEM, as shown in Fig. 6. Due to the washout effect of the heavy rain, the particles collected in the typical clear period of EP-2 were much smaller in size (Figs. 6a, b), which was in good agreement with the larger Ångström exponent. The coarse particles, such as dusts, were hardly observed, whereas a few K- rich particles were detected, of which presented in small cubic shape. Such particles could be explained by the coal combustion around the sampling site due to the slight fire spots presence. Besides, the cubic shape of K-rich particles suggested they have not undergone long transportation or severe photochemical reaction because cubic K-rich particles were generally generated from the molten nature of the material at high temperatures (Ault et al., 2012). Likewise, soot was generally less oxidized in the EP2 periods, maintaining fractional morphologies and externally mixed. Small metal particles and amorphous Zn-particles dominated the fine particles, which was ascribed to the industrial activity and/or waste incineration (Choël et al.,

2006;Moffet et al., 2008).

In the EP-5 episode, the increased aerosol loading played a remarkable role in the enhancement of scattering coefficient and decrease of visibility (Kang et al., 2013;Charlson et al., 1987;Deng et al., 2008).

Because of the high rate of aerosol collision, particles were larger than those in the clear days (Figs. 6c, d), leading to a smaller Ångström exponent. Almost of the soot particles observed under the TEM were compact and adhesive. It was internally mixed with the K-rich particles, which were larger, rounder or with a coating of high S components. As discussed above, they were probably transported from the south crop residual burning and undergo the ageing in the atmosphere, confirmed by the trajectories passing through intense fire spots. Due to the high concentration of soot, EP-5 were characterized by a high absorption coefficient, shown in Fig. 3.

The BC variations in the different weather types during the sampling period were illustrated in Fig. 7.

The preliminary component of BC could be viewed as the soot. High BC concentration was easily recognized in EP-5 with a mean value of 12.8 μg m$^{-3}$, while it is low up to 1.04 μg/m$^3$ during the clear periods. The former is about 11.3 times higher than that of the latter, which is due to the lower boundary layer In comparison, absorption coefficient of EP-5 (468.7 Mm$^{-1}$) was about 94.7 times higher than that of

EP-4 (1.3 Mm$^{-1}$), more than 8 times of the BC ratio. It was supposed that BC was internally mixed with other aerosols in the EP-5, which lead to the considerable elevation of absorption coefficient (Tan et al.,

2016). However, models estimated an enhancement of BC forcing up to a factor of 2.9 when BC is internally mixed with other aerosols, compared with externally mixed scenarios (Jacobson, 2001), which was much lower than this case. Accordingly, other light absorbing substances may contribute to the discrepancy. For example, brown carbon is an indispensable component of biomass burning, which has a strong absorption ability as well (Hoffer et al., 2006;Andreae and Gelencsér, 2006). Other particles such as dust may also contribute to the over-enhanced absorption coefficient (Yang et al., 2009). Our observations were agreement with the previous studies reported by (Wang et al., 2009b; Xia et al., 2006), which shows that aerosol particles under hazy weather conditions generate a positive heating effect on the atmospheric In the foggy days of EP-3 episode, the high PM$_{10}$ concentration and AOD caused significant increase of scattering coefficient (Tan et al., 2016). Furthermore, metal-bearing particles and soot were internally associated with some coatings including S-rich, N-rich and K-rich particles. Zhang et al. (2008) reported that coating with sulphuric acid enhance the optical properties of soot aerosols. Furthermore, the collected particles displayed larger size than those collected from the clear days under the TEM (Figs. 6e, f). The larger size particles in the foggy days could be caused by hydroscopic growth under the high relative humidity, and the collision among the overloading particles, which was likewise illustrated by the Ångström exponent shown in Fig.3.

Consequently, the larger particles enhance the scattering of sunlight, and lead to more apparent impairment of visibility (Quan et al., 2011). Chow et al. (2002a) reported that RH also has a profound impact on visibility. Some fan-like nitrate particles have inclusions which may act as the growth cores or be encompassed during the hydroscopic growth. Bian et al. (2009) reported that whenever the RH is elevated, its importance to AOD is substantially amplified if the particles are hygroscopic (Bian et al., 2009). Li et al. (2010) found that soot particles became hydrophilic when they were coated with the water-soluble compounds such as sulphates or nitrates, implying that soot can provide important nuclei for the development of aerosol particles. Furthermore, Fig. 6e and f illustrate a large fraction of internally mixed soot. It was not visible until being exposed to electron beam for a short time. As for an internally mixed particle, sulphate and nitrate coatings act as a "focusing mirror", and enhanced light absorption greatly.

Therefore, the BC concentration in foggy conditions was 6.12 $\mu g\, m^3$, and the absorption coefficient is 143.7

$Mm^{-1}$, which were 2.09 and 0.83 times of the hazy days, respectively. Model calculation also have recommended that light absorption ability of the internally mixed soot particles were enhanced by 30%

than that of soot alone (Fuller et al., 1999). A variety of metal particles were also observed in the foggy days, as foggy days had a stable low upper layer boundary and slight wind, leading to the accumulation of pollutions. These pollution sources range from steel plants and waste incineration to vehicle emission and so on (Hu et al., 2015).

**4 Conclusions**

The relationship between characteristics of aerosol particles and optical properties is of importance to the atmospheric chemistry research. However, the relationship between characteristics of aerosol particles and optical absorption remains poorly understood. Characteristics of aerosol optical properties, morphologies and their relationship were studied in urban Beijing during the clear, haze and fog episodes, sampled from 24th May to 22nd Jun, 2012. Transmission Electron Microscope (TEM), a Cavity Ring Down

Spectrometer (CRDS), a nephelometer and an aethalometer were employed to investigate the corresponding changes of the aerosol properties. Five episodes were categorised according to the meteorological conditions and composition. The results indicated that the clear episode (EP-2 and EP-4) was characterized as the low aerosol Optical Depth (AOD = 0.72) and less pollutants compared with haze (1.14) and fog (2.92)

episodes, which are mostly externally mixed. The high Ångström exponent (> 2.0) suggests that coarse particles were scarcely observed in EP-2 due to the washout of a previous heavy rain, whereas they were widespread in EP-4 (Ångström exponent = 0.04), which had some mineral particles introduced from the north. In contrast, industry-induced haze (EP-1) and biomass burning-induced haze (EP-5) were both affected by the south air mass. Higher AOD values illustrated heavy loading particle concentrations. All of the particles were classified into nine categories including S-rich, N-rich, mineral, K-rich, soot, tar ball, organic, metal and fly ash based on the TEM analysis. In the haze episode, as the influence of severe crop residue combustion, a large fraction of soot was detected, which sticks to sulphate or nitrate particles transformed from KCl. Both black carbon (BC) acceleration, internally mixed effects, and other light absorbing substances, contributed the light absorption enhancement. For foggy days, soot was mostly internally mixed with sulphates and nitrates, which revealed themselves after electron exposure under the

TEM. The larger size distribution was likely to be caused by both hygroscopic growth and collision between particles during the aging. About 28% of particles were internally mixed in the foggy days, which favored the light absorption. The comparison of all the episodes provides a deeper insight of how mixing states influence the aerosol extinction properties and also a clue to the air pollution control in the crop burning seasons. The result presented herein is beneficial to air pollution control and prevention in China.

**ACKNOWLEDGMENTS**

This work was supported by National Natural Science Foundation of China (Nos. 21577022, 21190053,

40975074), Ministry of Science and Technology of China (2016YFC0202700), and International cooperation project of Shanghai municipal government (15520711200).

[revised manuscript text omitted]

                                                               Figure 4

[Figure]

**Figure 5**

[Figure]

                                                    **Figure 6**

[Figure]

                                                                          **Figure 7**

---

## Author Comment (AC6) · 6 Mar 2017

Firstly, we acknowledge the comments of anonymous reviewers, and are also grateful to the efficient serving of the editor. We have already revised MS based on the reviewers' comments. We also inspected MS roundly and corrected some errors in English presentation. We are sure that the revised MS adhere to Atmospheric Chemistry and Physics format. The marked MS was also uploaded to be easily reviewed. Comments: 1. The English thorough the whole manuscript should be improved by a native English speaker. Response: The English thorough the manuscript have been improved by an English native speaker. Comments: 2. More recent reports about the influence of agricultural activities including biomass burning on the regional air quality are encouraged to be citied. Response: Many new published paper have been added into the manuscript. Comments: 3. Abstract: Line 12-16, the sentences are suggested to be replaced by "Aerosol optical properties and morphologies were measured by TEM, CRDS, a nephelometer and an aethalometer in a urban site of Beijing from 24 May to 22 June". The clear, haze and fog episodes just occurred during the sampling period, it didn't need to mention them in the abstract. The phrase of "sampled from..." is not correct in English grammar. The instruments were used for measuring the aerosol properties, but not for investigating the corresponding changes of the aerosol properties. Line 16- 17, the sentence is meaningless, because the individual episode was mentioned in the following sentences. Line 17-18, the phrase of "which are mostly externally mixed" is not clear, and hence suggested to be changed as "and the particles were mostly mostly externally mixed". Line 20-21, the comma before which should be deleted, because the phrase is used for modifying the EP-4. Line 21-22, "industry-induced haze (EP-1) and biomass burning-induce haze (EP-5)" is suggested to be changed as "the industryinduced pollution episode (EP-1) and biomass burning-induce pollution episode (EP-5) Line 22-24, The two sentences seemed to be independent, lack of logic, and thus, the two sentences are suggested to be replaced by "Compared with the EP-2 and EP-4, the AOD values and the size distribution of particles during EP-1 and EP-5 were much greater because of relatively high particle concentrations ". Line25-26, the sentence was suggested to be replaced by "In contrast to the EP-1, a large fraction of soot which sticks to KCl, sulphate or nitrate particles was detected during the EP-5", implying the evident influence of severe crop residue combustion. Line 26-28, the sentence was suggested to be replaced by "Additionally, evident enhancement of light absorption was observed during the EP-5, which was mainly ascribed to both BC acceleration and other absorbing substances". Line 28-31, the sentences are better replaced by "However, soot was found mostly internally mixed with sulphate and nitrate during a soot fog episode (EP-3), resulting in evident enhancement of light absorption". Response: Line 16: The sentences have been changed into "Aerosol optical properties and morphologies were measured by TEM, CRDS, a nephelometer and an

aethalometer in a urban site of Beijing from 24 May to 22 June". Line 17: Indeed, the sentence is meaningless and has been deleted. Line 23:"which" was replaced by "and the particles". The comma has been deleted Line 27: the sentences has been changed into "industry-induced haze (EP-1) and biomass burning-induced haze (EP-5) were both affected by the south air mass". Line 27-28: The two sentences have been replaced by "Compared with the EP-2 and EP-4, the AOD values and the size distribution of particles during EP-1 and EP-5 were much greater because of relatively high particle concentrations." Line 32-35: The sentence was replaced by "In contrast to the EP-1, a large fraction of soot which sticks to KCl, sulphate or nitrate particles was detected during the EP-5". Line 35: The sentence was replaced by "Additionally, evident enhancement of light absorption was observed during the EP-5, which was mainly ascribed to both BC acceleration and other absorbing substances". Line 39: The sentence has been changed into "However, soot was found mostly internally mixed with sulphate and nitrate during a soot fog episode (EP-3), resulting in evident enhancement of light absorption". Comments: 4. Line 44-47, any kind of particles in the atmosphere have scattering effect, why did you only stress on inorganic salts and light-color organic carbon? The sentences is better replaced by "inorganic salts and light-color organic carbon have a "cooling effect" on climate due to decreasing permeation of solar irradiation onto the Earth's surface through solely scattering sun light". There are still some sentences in the section being needed to be improved. Response: Line 57-58: Indeed, the sentence has been changed into "inorganic salts and light-color organic carbon have a cooling effect". Comments: 5. Line 266-267, the sentence of "the north wind was relatively clean and the time was insufficient for a heavy accumulation" is not proper and clear. Wind can be only described by speed and direction, and hence the sentence is better replaced by "the air parcel from the North was relatively clean". What's "the time" in the latter half sentence"? Line 282, the title of "Optical parameter variation" is better replaced by "The variation of aerosol optical characters". Line 324, the title of "TEM analysis" is suggested to be "Morphology and chemical composition of aerosols". Line 319: "Especially when air masses moved from south direction to

the sampling site aerosols were influenced by heavy soot-sulfate-OC-mixed pollution". How did you draw the conclusion? Line 388: Are you sure haze and fog episodes had a high possibility of collision just due to the heavy particle loading? You should add relevant reference to confirm your deduction. Line 390, the title of "Optical properties related to morphological of aerosols" is better replaced by "the relationship between the aerosol optical properties and morphologies". Response: Line 280: The sentence was changed into "the air parcel from the North was relatively clean". Line 295: The sentence was changed into "The variation of aerosol optical characters" Title 3.3: The title was replaced by "Morphology and chemical composition of aerosols" Line 334: The conclusion was drawed because some previous studies have confirmed that soot, organic matter, and sulfates were generated from the industrial activities, domestic cooking, and biomass burning. Many industrial activities and biomass burning have been observed in South China. Line 404: Haze and fog episodes generally had a high possibility of collision, which was caused by heavy particle loading. In addition, prolonged remaining of heavy particles was also a factor leading to the collision. Many relevant references have been added in the manuscript. Line 407: The title was replaced by "the relation of optical properties and the morphologies of aerosol particles".

---

## Author Comment (AC7) · 6 Mar 2017

Dear editor: Here we submit our revised manuscript for consideration to be published on Atmospheric chemistry and physics. The further information about our manuscript is as follows: Topic: Real-Time Aerosol Optical Properties, Morphology and Mixing States under Clear, Haze and Fog Episodes in the summer of Urban Beijing Type of Manuscript: article Authors: Rui Li1, Yunjie Hu1, Ling Li1, Hongbo Fu1,2,*, Jianmin Chen1,* Corresponding author: Hongbo Fu; Address: Department of Environmental Science and Engineering, Fudan University, Shanghai 200433, China; Tel.: (+86)21-5566-5189; Fax: (+86)21-6564-2080; Email: fuhb@fudan.edu.cn. Jianmin Chen; Address: Department of Environmental Science and Engineering, Fudan Uni-

versity, Shanghai 200433, China; Tel.: (+86)21-5566-5189; Fax: (+86)21-6564-2080; Email: jmchen@fudan.edu.cn

Firstly, we acknowledge the comments of anonymous reviewers, and are also grateful to the efficient serving of the editor. We have already revised MS based on the reviewers' comments. We also inspected MS roundly and corrected some errors in English presentation. We are sure that the revised MS adhere to Atmospheric Chemistry and Physics format. The marked MS was also uploaded to be easily reviewed. Comments: 1. The English thorough the whole manuscript should be improved by a native English speaker. Response: The English thorough the manuscript have been improved by an English native speaker. Comments: 2. More recent reports about the influence of agricultural activities including biomass burning on the regional air quality are encouraged to be citied. Response: Many new published paper have been added into the manuscript. Comments: 3. Abstract: Line 12-16, the sentences are suggested to be replaced by "Aerosol optical properties and morphologies were measured by TEM, CRDS, a nephelometer and an aethalometer in a urban site of Beijing from 24 May to 22 June". The clear, haze and fog episodes just occurred during the sampling period, it didn't need to mention them in the abstract. The phrase of "sampled from..." is not correct in English grammar. The instruments were used for measuring the aerosol properties, but not for investigating the corresponding changes of the aerosol properties. Line 16- 17, the sentence is meaningless, because the individual episode was mentioned in the following sentences. Line 17-18, the phrase of "which are mostly externally mixed" is not clear, and hence suggested to be changed as "and the particles were mostly mostly externally mixed". Line 20-21, the comma before which should be deleted, because the phrase is used for modifying the EP-4. Line 21-22, "industry-induced haze (EP-1) and biomass burning-induce haze (EP-5)" is suggested to be changed as "the industryinduced pollution episode (EP-1) and biomass burning-induce pollution episode (EP-5) Line 22-24, The two sentences seemed to be independent, lack of logic, and thus, the two sentences are suggested to be replaced by "Compared with the EP-2 and EP-4, the AOD values and the size distribution of particles during EP-1 and EP-5 were much greater because of relatively high particle concentrations ". Line25-26, the sentence was suggested to be replaced by "In contrast to the EP-1, a large fraction of soot which sticks to KCl, sulphate or nitrate particles was detected during the EP-5", implying the evident influence of severe crop residue combustion. Line 26-28, the sentence was suggested to be replaced by "Additionally, evident enhancement of light absorption was observed during the EP-5, which was mainly ascribed to both BC acceleration and other absorbing substances". Line 28-31, the sentences are better replaced by "However, soot was found mostly internally mixed with sulphate and nitrate during a soot fog episode (EP-3), resulting in evident enhancement of light absorption". Response: Line 16: The sentences have been changed into "Aerosol optical properties and morphologies were measured by TEM, CRDS, a nephelometer and an aethalometer in a urban site of Beijing from 24 May to 22 June". Line 17: Indeed, the sentence is meaningless and has been deleted. Line 23:"which" was replaced by "and the particles". The comma has been deleted Line 27: the sentences has been changed into "industry-induced haze (EP-1) and biomass burning-induced haze (EP-5) were both affected by the south air mass". Line 27-28: The two sentences have been replaced by "Compared with the EP-2 and EP-4, the AOD values and the size distribution of particles during EP-1 and EP-5 were much greater because of relatively high particle concentrations." Line 32-35: The sentence was replaced by "In contrast to the EP-1, a large fraction of soot which sticks to KCl, sulphate or nitrate particles was detected during the EP-5". Line 35: The sentence was replaced by "Additionally, evident enhancement of light absorption was observed during the EP-5, which was mainly ascribed to both BC acceleration and other absorbing substances". Line 39: The sentence has been changed into "However, soot was found mostly internally mixed with sulphate and nitrate during a soot fog episode (EP-3), resulting in evident enhancement of light absorption". Comments: 4. Line 44-47, any kind of particles in the atmosphere have scattering effect, why did you only stress on inorganic salts and light-color organic carbon? The sentences is better replaced by "inorganic salts and light-color organic carbon have a "cooling effect" on climate due to decreasing permeation of solar irradiation onto the Earth's surface through solely scattering sun light". There are still some sentences in the section being needed to be improved. Response: Line 57-58: Indeed, the sentence has been changed into "inorganic salts and light-color organic carbon have a cooling effect". Comments: 5. Line 266-267, the sentence of "the north wind was relatively clean and the time was insufficient for a heavy accumulation" is not proper and clear. Wind can be only described by speed and direction, and hence the sentence is better replaced by "the air parcel from the North was relatively clean". What's "the time" in the latter half sentence"? Line 282, the title of "Optical parameter variation" is better replaced by "The variation of aerosol optical characters". Line 324, the title of "TEM analysis" is suggested to be "Morphology and chemical composition of aerosols". Line 319: "Especially when air masses moved from south direction to the sampling site aerosols were influenced by heavy soot-sulfate-OC-mixed pollution". How did you draw the conclusion? Line 388: Are you sure haze and fog episodes had a high possibility of collision just due to the heavy particle loading? You should add relevant reference to confirm your deduction. Line 390, the title of "Optical properties related to morphological of aerosols" is better replaced by "the relationship between the aerosol optical properties and morphologies". Response: Line 280: The sentence was changed into "the air parcel from the North was relatively clean". Line 295: The sentence was changed into "The variation of aerosol optical characters" Title 3.3: The title was replaced by "Morphology and chemical composition of aerosols" Line 334: The conclusion was drawed because some previous studies have confirmed that soot, organic matter, and sulfates were generated from the industrial activities, domestic cooking, and biomass burning. Many industrial activities and biomass burning have been observed in South China. Line 404: Haze and fog episodes generally had a high possibility of collision, which was caused by heavy particle loading. In addition, prolonged remaining of heavy particles was also a factor leading to the collision. Many relevant references have been added in the manuscript. Line 407: The title was replaced by "the relation of optical properties and the morphologies of aerosol particles".

Please also note the supplement to this comment:
http://www.atmos-chem-phys-discuss.net/acp-2016-976/acp-2016-976-AC7-supplement.pdf

**Supplement:**

**Real-Time Aerosol Optical Properties, Morphology and Mixing**

**States under Clear, Haze and Fog Episodes in the summer of Urban**

**Beijing**

Rui Li[1], Yunjie Hu[1], Ling Li[1], Hongbo Fu[1,2,*], Jianmin Chen[1,*]

*[1] Shanghai Key Laboratory of Atmospheric Particle Pollution and Prevention, Department of*

*Environmental Science & Engineering, Fudan University, Shanghai 200433, China Chinese*

*Academy of Sciences, Institute of Atmosphere Physics, Beijing 100029.*

*[2] Collaborative Innovation Center of Atmospheric Environment and Equipment Technology*

*(CICAEET), Nanjing University of Information Science and Technology, Nanjing 210044,*

*China*

**Abstract**

Aerosol particles play significant roles on the climate-forcing agent via its optical absorption properties. However, the relationship between characteristics of aerosol particles and optical absorption remains poorly understood. Aerosol optical properties and morphologies were measured by TEM, CRDS, a nephelometer and an aethalometer in a urban site of Beijing from

24 May to 22 June.

nd

Five episodes were categorised according to the meteorological conditions and composition. The results showed that the clear episode (EP-2 and EP-4) featured as the low Aerosol Optical Depth (AOD = 0.72) and less pollutants compared with haze (1.14) and fog (2.92) episodes and the particles,  are mostly externally mixed. The high Ångström exponent (> 2.0) suggests that coarse particles were scarcely observed in EP-2 due to the washout of a previous heavy rain, whereas they were widespread in EP-4 (Ångström exponent = 0.04), which had some mineral particles introduced from the north. In contrast, industry-induced haze (EP-1) and biomass burning-induced haze (EP-5) were both affected by the south air mass. Compared with the EP-2 and EP-4, the AOD

values and the size distribution of particles during EP-1 and EP-5 were much greater because of relatively high particle concentrations. All of the particles were classified into nine categories including S-rich, N-rich, mineral, K-rich, soot, tar ball, organic, metal and fly ash on the basis of TEM analysis. In the haze episode, In contrast to the EP-1, a large fraction of soot, which sticks to KCl, sulphate or nitrate particles was detected during the EP-5

Additionally, evident enhancement of light absorption was observed during the EP-5, which was mainly ascribed to both BC acceleration and other absorbing substances.

However, soot was found mostly internally mixed with sulphate and nitrate during a soot fog episode (EP-3), resulting in evident enhancement of light absorption

[revised manuscript text omitted]
, suggesting that more soot is uploaded into the atmosphere during this period. It is well known that soot emission is much higher in the past years, mainly contributed by the residential coal combustion, biomass burning, coke production, and diesel vehicles (Wang et al., 2012b). Especially, when air masses moved from south direction the sampling site were influenced by heavy polluted air mass mixed by soot, sulfate, and OC-components, from the dense population centres and industrial areas (Sun et al.,

2006;Wang et al., 2006), which was also confirmed by the TEM observation.

**3.3**  “**Morphology and chemical composition of aerosols**

Based on morphology and chemical composition, 1173 particles were classified into nine categories: S-rich (Fig. 4a), N-rich (Fig. 4b), mineral (Fig. 4c), K-rich (Fig. 4d), soot (Fig. 4e), tar ball (Fig.4f), organic (Fig.4g), metal (Fig. 4h) and fly ash (Fig. 4i). The classification is similar to the work reported by Li and

Shao (2009).

The most common particles are sulphates and nitrates (Figs. 4a and b), which are of the size around 1.0

μm, and have a light scattering ability (Jacobson, 2001). Sulphates appeared as subrounded masses under the TEM, which decomposed or evaporated under the electron beam exposure. Conventionally, they were formed by the reaction of precursor $SO_2$ or $H_2SO_4$ with other gases or particles (Khoder, 2002). Nitrates were mostly of scalloped morphology in the TEM images. They were relatively stable when exposed to the electron beam. Nitrates formed through the homogeneous reaction with the precursor either $NO_2$ or heterogenic reaction with $HNO_3$ (Khoder, 2002). (Pathak et al., 2004; Seinfeld and Pandis, 2012).

In the clear days, as the result of effects of northern air mass, dust particles were relatively abundant. The size of dust particles (Fig. 4c) were large, usually bigger than 1.0 μm, so far as to 8.0 μm. Their compositions differed from each other, mostly are silicates and calcium sulphate or carbonate, all of which were stable under the exposure of the electron beam. Dust particles were reported to have a light scattering effect, resulting in a negative aerosol radiative forcing (Wang et al., 2009b). They took up a large portion in EP-4, impacted by the north wind taking along particles from the dusty regions.

As for the haze episode, K-rich particles (Li et al., 2010; Duan et al., 2004; Engling et al., 2009), soot (Li et al., 2010), tar ball (Chakrabarty et al., 2010;Bond, 2001) and organic (Lack et al., 2012) were more observed under the TEM. K-rich particles (Fig. 4d) often existed as sulphate or nitrate. A larger fraction of

K-rich particles was observed in EP-5 than those in the other periods. Together with the back trajectories and fire spot maps, it was supposed that the regional haze occurred in EP-5 was contributed significantly by the biomass burning. K-rich particles were characterized by the irregular shape, which was unstable when exposed to electron beam. KCl was barely detected in the samples, even though it has been recommended that KCl was internally mixed with $K_2SO_4$ and $KNO_3$ in fresh biomass burning plumes (Li et al., 2010;Li et al., 2003;Adachi and Buseck, 2008). Based on the EDS data, K-rich particles in the present work mostly consisted of N, Na, O, S, and K, whereas it was free of Cl, implying KCl could have suffered from chemical reactions and transformed into sulphates or nitrates (Li and Shao, 2010). Such particles displayed a negative climate forcing (Hauglustaine et al., 2014).

It was well documented that soot (Fig. 4e) was vital to light absorption, which could alter regional atmospheric stability and vertical motions, the large scale circulation and precipitation with significant regional climate effects (Ramanathan et al., 2001;Jacobson, 2002). It was well characterized by a structure like onion ring, resembling a fractal long chain as agglomerates of small spherical monomers (Li and Shao,

2009). The fresh soot was loose and externally mixed. However, after undergoing a long-range transportation and aging in the atmosphere, soot became more compacted, with a slight increase of O

concentration because of the photochemistry (Stanmore et al., 2001; Krasowsky et al., 2016). Meanwhile, soot generally attached to other particles on the surface or serves as the core for other particle formation.

Tar ball (Fig. 4f) was present as a spherical carbon ball with a small fraction of O. It was thought to origin from the smouldering combustion and have relatively strong absorption effects (Chakrabarty et al.,

2010; Bond, 2001). Tar balls constituted a large fraction of the fresh emitted wildfire carbonaceous particles (China et al., 2013;Lack et al., 2012). But it was seldom observed in the present work, even in EP-5 when there was severe biomass burning emission, which may be due to the difference in burning species and conditions.

Organic matter (Fig. 4g) identified by HRTEM was amorphous species, and was stable under the strong electron beam exposure. It could be traced to the direct emission such as biomass burning (Lack et al.,

2012), or the second reaction between VOCs with ozone (Wang et al., 2012a). It can absorb radiation in the low-visible and UV wavelengths (Chakrabarty et al., 2010;Clarke et al., 2007;Lewis et al., 2008;Hoffer et al., 2006). In addition, when compassing soot as the core, organic matter can enhance absorption by internal mixing (Adachi and Buseck, 2008).

For the common haze and fog episodes, the stagnated weather favours the accumulation of pollutants, especially metal particles and fly ash (Hu et al., 2015). Metal particles (Fig. 4h) were generally round and stable under the TEM. Fly ash (Fig. 4i) was a dark sphere with large size of $> 1$ μm. It was a common product of industrial activities in the northern China (Shi et al., 2003). As the complex refractive index (CRI) indicated, metal oxide particles and fly ash can scatter light, but the former has a weak absorption ability while the later has almost no light absorption ability (Ebert et al., 2004).

Figure 5 shows percentage of nine components in clear, haze and fog episodes under external mixing, internal mixing and adjacent states (partially internal mixing). About 28% of particles were internally mixed in the foggy days, while about 52% of particles exhibited external mixing state in clear days based on the

TEM analysis. Mineral particles were inclined to be externally mixed with K-rich particles and organic matter in clear days, while the external ratio of other particles were relatively lower, particularly in the haze and fog days. Li et al. (2010) showed that mineral particles generally displayed external association with organic matter or other particles. However, many fine particles including metal-bearing particles, fly ash and soot were often internally mixed with S-rich and K-rich particles, particularly during the fog-haze episodes. Shi et al. (2008) reported that rapid aging of fresh soot tended to appear during the fog-haze days, which were generally associated with ammonium sulfate. Heavy polluted air generally promoted the coagulation between S/K-rich particles and those fine particles such as metal particles, soot, and fly ash (Li and Shao, 2009), which could explain the results. Additionally,  haze and fog episodes held a higher possibility of collision and attachment due to the heavy particle loading and prolonged remaining in the atmosphere (Li and Shao, 2009; Li et al., 2010), leading to a higher internal mixed state percentage around

65%.

**3.4 The relation of optical properties and the morphologies of aerosol particles**

The different morphologies of the particles collected from the different weather can be easily identified under the TEM, as shown in Fig. 6. Due to the washout effect of the heavy rain, the particles collected in the typical clear period of EP-2 were much smaller in size (Figs. 6a, b), which was in good agreement with the larger Ångström exponent. The coarse particles, such as dusts, were hardly observed, whereas a few K- rich particles were detected, of which presented in small cubic shape. Such particles could be explained by the coal combustion around the sampling site due to the slight fire spots presence. Besides, the cubic shape of K-rich particles suggested they have not undergone long transportation or severe photochemical reaction because cubic K-rich particles were generally generated from the molten nature of the material at high temperatures (Ault et al., 2012). Likewise, soot was generally less oxidized in the EP2 periods, maintaining fractional morphologies and externally mixed. Small metal particles and amorphous Zn-particles dominated the fine particles, which was ascribed to the industrial activity and/or waste incineration (Choël et al.,

2006;Moffet et al., 2008).

In the EP-5 episode, the increased aerosol loading played a remarkable role in the enhancement of scattering coefficient and decrease of visibility (Kang et al., 2013;Charlson et al., 1987;Deng et al., 2008).

Because of the high rate of aerosol collision, particles were larger than those in the clear days (Figs. 6c, d), leading to a smaller Ångström exponent. Almost of the soot particles observed under the TEM were compact and adhesive. It was internally mixed with the K-rich particles, which were larger, rounder or with a coating of high S components. As discussed above, they were probably transported from the south crop residual burning and undergo the ageing in the atmosphere, confirmed by the trajectories passing through intense fire spots. Due to the high concentration of soot, EP-5 were characterized by a high absorption coefficient, shown in Fig. 3.

The BC variations in the different weather types during the sampling period were illustrated in Fig. 7.

The preliminary component of BC could be viewed as the soot. High BC concentration was easily recognized in EP-5 with a mean value of 12.8 μg m$^{-3}$, while it is low up to 1.04 μg/m$^3$ during the clear periods. The former is about 11.3 times higher than that of the latter, which is due to the lower boundary layer. In comparison, absorption coefficient of EP-5 (468.7 Mm$^{-1}$) was about 94.7 times higher than that of

EP-4 (1.3 Mm$^{-1}$), more than 8 times of the BC ratio. It was supposed that BC was internally mixed with other aerosols in the EP-5, which lead to the considerable elevation of absorption coefficient (Tan et al.,

2016). However, Models models estimated an enhancement of BC forcing up to a factor of 2.9 when BC is internally mixed with other aerosols, compared with externally mixed scenarios (Jacobson, 2001), which was much lower than this case. Accordingly, other light absorbing substances may contribute to the discrepancy. For example, Brown brown carbon is an indispensable component of biomass burning, which has a strong absorption ability as well (Hoffer et al., 2006;Andreae and Gelencsér, 2006). Other particles like such as dust may also contribute to the over-enhanced absorption coefficient (Yang et al., 2009). Our observations were agreement with the previous studies reported by (Wang et al., 2009b; Xia et al., 2006), which shows that aerosol particles under hazy weather conditions generate a positive heating effect on the atmospheric column (Wang et al., 2009b;Xia et al., 2006).

In the foggy days of EP-3 episode, the high PM$_{10}$ concentration and AOD caused significant increase of scattering coefficient (Tan et al., 2016). Furthermore, metal-bearing particles and soot were internally associated with some coatings including S-rich, N-rich and K-rich particles. Zhang et al. (2008) reported that coating with sulphuric acid enhance the optical properties of soot aerosols. Furthermore, the collected particles displayed larger size than those collected from the clear days under the TEM (Figs. 6e, f). The larger size particles in the foggy days could be caused by hydroscopic growth under the high relative humidity, and the collision among the overloading particles, which was likewise illustrated by the Ångström exponent shown in Fig.3. Consequently, the larger particles enhance the scattering of sunlight, and lead to more apparent impairment of visibility (Quan et al., 2011). Chow et al. (2002a) reported that RH also has a profound impact on visibility. Some fan-like nitrate particles have inclusions which may act as the growth

cores or be encompassed during the hydroscopic growth. Bian et al. (2009) reported that whenever the RH

is elevated, its importance to AOD is substantially amplified if the particles are hygroscopic (Bian et al.,

2009). Li et al. (2010) found that soot particles became hydrophilic when they were coated with the water- soluble compounds such as sulphates or nitrates, implying that soot can provide important nuclei for the development of aerosol particles. Furthermore, Fig. 6e and f illustrate a large fraction of internally mixed soot. It was not visible until being exposed to electron beam for a short time. As for an internally mixed particle, sulphate and nitrate coatings act as a "focusing mirror", and enhanced light absorption greatly.

Therefore, the BC concentration in foggy conditions was 6.12 $\mu g\,m^3$, and the absorption coefficient is 143.7

$Mm^{-1}$, which were 2.09 and 0.83 times of the hazy days, respectively. Model calculation also have recommended that light absorption ability of the internally mixed soot particles were enhanced by 30%

than that of soot alone (Fuller et al., 1999). A variety of metal particles were also observed in the foggy days, as foggy days had a stable low upper layer boundary and slight wind, leading to the accumulation of pollutions. These pollution sources range from steel plants and waste incineration to vehicle emission and so on (Hu et al., 2015).

**4 Conclusions**

The relationship between characteristics of aerosol particles and optical properties is of importance to the atmospheric chemistry research. However, the relationship between characteristics of aerosol particles and optical absorption remains poorly understood. Characteristics of aerosol optical properties, morphologies and their relationship were studied in urban Beijing during the clear, haze and fog episodes, sampled from 24th May to 22nd Jun, 2012. Transmission Electron Microscope (TEM), a Cavity Ring Down

Spectrometer (CRDS), a nephelometer and an aethalometer were employed to investigate the corresponding changes of the aerosol properties. Five episodes were categorised according to the meteorological conditions and composition. The results indicated that the clear episode (EP-2 and EP-4) was characterized as the low aerosol Optical Depth (AOD = 0.72) and less pollutants compared with haze (1.14) and fog (2.92)

episodes, which are mostly externally mixed. The high Ångström exponent (> 2.0) suggests that coarse particles were scarcely observed in EP-2 due to the washout of a previous heavy rain, whereas they were widespread in EP-4 (Ångström exponent = 0.04), which had some mineral particles introduced from the north. In contrast, industry-induced haze (EP-1) and biomass burning-induced haze (EP-5) were both affected by the south air mass. Higher AOD values illustrated heavy loading particle concentrations. All of the particles were classified into nine categories including S-rich, N-rich, mineral, K-rich, soot, tar ball, organic, metal and fly ash based on the TEM analysis. In the haze episode, as the influence of severe crop residue combustion, a large fraction of soot was detected, which sticks to sulphate or nitrate particles transformed from KCl. Both black carbon (BC) acceleration, internally mixed effects, and other light absorbing substances, contributed the light absorption enhancement. For foggy days, soot was mostly internally mixed with sulphates and nitrates, which revealed themselves after electron exposure under the TEM. The larger size distribution was likely to be caused by both hygroscopic growth and collision between particles during the aging. About 28% of particles were internally mixed in the foggy days, which favored the light absorption. The comparison of all the episodes provides a deeper insight of how mixing states influence the aerosol extinction properties and also a clue to the air pollution control in the crop burning seasons. The result presented herein is beneficial to air pollution control and prevention in China.

**ACKNOWLEDGMENTS**

[revised manuscript text omitted]

Takemura, T., Nakajima, T., Dubovik, O., Holben, B. N., and Kinne, S.: Single-scattering albedo and radiative forcing of various aerosol species with a global three-dimensional model, J. Climate., 15, 333-

352, 2002.

Tan, H.B., Liu, L., Fan, S.J., Li, F., Yin, Y., Cai, M.F., Chan, P.W., 2016. Aerosol optical properties and mixing state of black carbon in the Pearl River Delta, China, Atmos. Environ., 131, 196-208, 2016.

[revised manuscript text omitted]

                                         Figure 3

[Figure]

                              **Figure 4**

[Figure]

                          **Figure 5**

[Figure]

**Figure 6**

[Figure]

Figure 7